# A population of innate myelolymphoblastoid effector cell expanded by inactivation of mTOR complex 1 in mice

**Fei Tang[1], Peng Zhang[1,2], Peiying Ye[1], Christopher A Lazarski[1], Qi Wu[3], Ingrid L Bergin[4], Timothy P Bender[5], Michael N Hall[6], Ya Cui[2], Liguo Zhang[7], Taijiao Jiang[2], Yang Liu[1]\*, Pan Zheng[1]\***

[1]Center for Cancer and Immunology Research, Children's Research Institute, Children's National Medical Center, Washington, United States; [2]Key Laboratory of Protein and Peptide Pharmaceuticals, Institute of Biophysics, Chinese Academy of Sciences, Beijing, China; [3]Department of Neurology, University of Michigan Medical School, Ann Arbor, United States; [4]ULAM In-Vivo Animal Core, University of Michigan Medical School, Ann Arbor, United States; [5]Department of Microbiology, Immunology and Cancer Biology, University of Virginia, Charlottesville, United States; [6]Biozentrum, University of Basel, Basel, Switzerland; [7]Key Laboratory of Infection and Immunity, Institute of Biophysics, Chinese Academy of Sciences, Beijing, China

**\*For correspondence:**
yaliu@cnmc.org (YL);
pzheng@cnmc.org (PZ)

**Competing interests:** The authors declare that no competing interests exist.

**Abstract** Adaptive autoimmunity is restrained by controlling population sizes and pathogenicity of harmful clones, while innate destruction is controlled at effector phase. We report here that deletion of *Rptor* in mouse hematopoietic stem/progenitor cells causes self-destructive innate immunity by massively increasing the population of previously uncharacterized innate myelolymphoblastoid effector cells (IMLECs). Mouse IMLECs are CD3⁻B220⁻NK1.1⁻Ter119⁻ CD11c$^{low/-}$CD115⁻F4/80$^{low/-}$Gr-1⁻ CD11b⁺, but surprisingly express high levels of PD-L1. Although they morphologically resemble lymphocytes and actively produce transcripts from Immunoglobulin loci, IMLECs have non-rearranged *Ig* loci, are phenotypically distinguishable from all known lymphocytes, and have a gene signature that bridges lymphoid and myeloid leukocytes. *Rptor* deletion unleashes differentiation of IMLECs from common myeloid progenitor cells by reducing expression of *Myb*. Importantly, IMLECs broadly overexpress pattern-recognition receptors and their expansion causes systemic inflammation in response to Toll-like receptor ligands in mice. Our data unveil a novel leukocyte population and an unrecognized role of Raptor/mTORC1 in innate immune tolerance.
DOI: https://doi.org/10.7554/eLife.32497.001

## Introduction

Autoreactive T and B lymphocytes are controlled by regulation of population sizes and pathogenicity through clonal deletion (*Burnet, 1957*), clonal anergy (*Nossal and Pike, 1980*) and regulatory T cells (*Sakaguchi et al., 1995*). Through broadly-reactive pattern recognition receptors (PRRs) (*Janeway, 1989*; *Medzhitov et al., 1997*), innate immunity protects host against infections by both direct effector function and, indirectly, by induction of adaptive immunity (*Liu and Janeway, 1991*; *Liu and Janeway, 1992*; *Wu and Liu, 1994*; *Pasare and Medzhitov, 2005*). Since innate immune responses triggered by host components can also cause fatal tissue damage (*Stetson et al., 2008*;

**eLife digest** The cells of the immune system defend us from bacteria, viruses and other microbes that might cause harm to the body. Immune cells develop from stem cells in the bone marrow. The stem cells first develop into one of two types of progenitor cell before specializing further into the different types of mature immune cell. Researchers often categorize immune cells as either myeloid or lymphoid, depending on which progenitor cell they developed from.

A protein complex called mTORC1 in the stem cells helps to guide immune cell development. One of the proteins in the mTORC1 complex is called Raptor. In mice that lack the Raptor protein, cells with particular markers on their surface accumulate in the bone marrow. The exact identity of these cells and why they appear was not known.

Tang et al. have analyzed these cells in mice that lacked Raptor in their bone marrow stem cells. This revealed that the cells have features of both myeloid and lymphoid cells: although they develop from myeloid progenitor cells, they are shaped like lymphoid cells. The cells also have a surface marker normally found on myeloid cells. Tang et al. have named the cells innate myelolymphoblastoid effector cells (IMLECs).

Further investigation showed that the lack of the Raptor protein caused another gene in the stem cells, called *Myb*, to become less active than normal. Tang et al. suggest that this lack of activity causes more IMLECs to develop from the stem cells. The overproduction of IMLECs also causes inflammation in the mice.

IMLECs bridge the gap between myeloid and lymphoid cells, challenging the current categorization of these as separate cell types. Targeting this new cell population could help researchers to develop new methods to control the immune response, for example, during autoimmune disorders where the immune system is overactive and damages the body's own cells.

DOI: https://doi.org/10.7554/eLife.32497.002

*Chen et al., 2009*), it must be properly regulated to protect hosts against self-destruction. Although a number of mechanisms have been proposed to prevent self-destructive innate effector functions (*Liu et al., 2009*; *Liu et al., 2011*; *Ljunggren and Kärre, 1990*; *Liew et al., 2005*; *Takeuchi and Akira, 2010*), it is less clear if population sizes of innate effectors are suppressed to limit self-destruction.

The significance of protective self-tolerance mechanisms in adaptive immunity are revealed only when they have gone awry. For example, the significance of clonal deletion was elucidated when it was prevented by blocking either costimulation or antigen-expression in the thymus (*Anderson et al., 2002*; *Gao et al., 2002*), while mice with Foxp3 mutation informed us of the consequence of defective regulatory T cells (*Hori et al., 2003*; *Fontenot et al., 2003*). Since removal of the forbidden autoreactive T and B cells is achieved during T and B cell development (*Nemazee and Bürki, 1989*; *Sha et al., 1988*; *Kisielow et al., 1988*), it is intriguing whether certain parallel mechanisms in innate immunity might also remain to be uncovered through genetic inactivation of key regulators in development of innate effector cells.

The hematopoietic system is among the best characterized of all tissue/systems in mammalians, with cell types and lineages clearly defined in the context of developmental stages and localization (*Metcalf, 1999*; *Weissman, 1994*). Hematopoiesis in bone marrow (BM) is responsible for generation of major lineages of innate effectors, including NK cells, granulocytes, monocytes and dendritic cells. While genetic switch in generation of innate immune system has been identified (*Weissman, 1994*), we are not aware of defects that predispose host to innate immune attack through increasing population sizes of self-destructive innate effectors.

The mammalian target of rapamycin (mTOR) pathway, which couples energy and nutrient abundance to the execution of cell growth and division, has emerged as a major regulator of hematopoiesis. Thus, activation of mTOR complex 1 (mTORC1) by deleting *Tsc1*, which encodes a negative regulator for mTORC1 (*Inoki et al., 2002*), causes loss of hematopoietic stem cell (HSC) function and renders mice prone to leukemiogenesis in conjunction with loss of tumor suppressor *Pten* (*Yilmaz et al., 2006*; *Zhang et al., 2006*; *Chen et al., 2008*). More recently, two groups reported that deletion of *Rptor*, which encodes a critical component of mTORC1 (*Hara et al., 2002*;

*Kim et al., 2002*), dramatically perturbed hematopoiesis in mice (*Hoshii et al., 2012*; *Kalaitzidis et al., 2012*), as evidenced by defects in production of mature lymphoid and myeloid cells. Remarkably, cells with CD11b$^+$ Gr-1$^-$ surface markers massively accumulated in BM following *Rptor* deletion in HSCs (*Hoshii et al., 2012*; *Kalaitzidis et al., 2012*). The nature of this population and consequences of their accumulation, however, remains a mystery.

Here we systematically analyzed the gene expression signature, cell surface markers, morphology and functions of the CD11b$^+$Gr-1$^-$ population in the *Rptor*-deficient BM and other organs and sought for their physiological counterpart in the normal mice. We found that these cells can be identified in both normal and *Rptor*-deficient hosts by CD3$^-$B220$^-$NK1.1$^-$Ter119$^-$ CD11c$^{low/-}$CD115$^-$F4/80$^{low/-}$Gr-1$^-$ CD11b$^+$PD-L1$^+$ markers, lymphoid morphology and actively transcribed *Ig* loci. Interestingly, these cells broadly express essentially all TLRs along with many other pattern recognition receptors and mount a greatly exacerbated response to all TLR ligands tested. We name this population IMLEC for innate myelolymphoblastoid effector cell that can be derived from common myeloid progenitors. Because their expansion and broad distribution render the host vulnerable to TLR ligands, we suggest that mTORC1-mediated repression of IMLEC expansion represents a new mechanism of immune tolerance in the innate immunity. Our study also raises an intriguing perspective that while repressing mTOR over-activation suppresses leukemia, a functional mTORC1 must be maintained to limit generation of IMLECs to avoid innate immune destruction.

## Results

### Raptor suppresses accumulation of a previously uncharacterized subset of leukocytes with features of both myeloid and lymphoid cells

As germline deletion of *Rptor* (which encode the Raptor protein) is embryonic-lethal, we crossed mice harboring homozygous loxp-flanked *Rptor* exon 6 (*Polak et al., 2008*) to those with interferon-inducible *Mx1-Cre* recombinase transgene, which allows inducible deletion of target genes effectively in the hematopoietic system upon treatment of interferon or its inducers (*Kühn et al., 1995*). We treated the 6–8 weeks old *Rptor* $^{F/F}$ and *Rptor* $^{F/F}$,*Mx1-Cre* mice with polyinosinic: polycytidylic acid (pIpC) every other day for 2 weeks to induce the deletion of *Rptor*. Hereafter, we refer to the pIpC-treated *Rptor* $^{F/F}$ mice as Ctrl (control) mice, while the *Rptor* $^{F/F}$, *Mx1-Cre* mice as cKO (conditional knockout) mice (*Figure 1A* and *Figure 1—figure supplement 1*). As has been reported by others (*Hoshii et al., 2012*; *Kalaitzidis et al., 2012*), *Rptor* deletion causes broad defects in all lineages of hematopoietic cells (see also *Figure 1—figure supplements 1*, *2* and *3*). However, the number of hematopoietic stem/progenitor cells (HSPCs) increased (*Figure 1—figure supplement 4*). Most notably, CD11b$^+$ Gr-1$^-$ cells, which amount to nearly 50% of BM cells in our model, emerge at the expense of CD11b$^+$ Gr-1$^+$ granulocytes from the cKO mice (*Figure 1B,C*). Importantly, we also observed the massive accumulation of CD11b$^+$Gr-1$^-$ cells in the BM of *Rptor$^{F/F}$*, *Cre-ER* mice after tamoxifen induced targeted mutation of *Rptor*, which clearly excludes the role of pIpC in the generation of these cells (*Figure 1—figure supplement 5*).

The CD11b$^+$ Gr-1$^-$ cells were smaller and had reduced granularity when compared to CD11b$^+$ Gr-1$^+$ granulocytes, but were larger and more granular than the CD11b$^-$ Gr-1$^-$ cells (*Figure 1—figure supplement 2E,F*). Surprisingly, despite the expression of myeloid marker CD11b on the expanded population of BM cells, histological analysis of BM section revealed pervasive expansion of lymphoblastoid cells (*Figure 1D*). The cKO BM contained markedly decreased erythroid and myeloid lineage cells and markedly increased lymphocytes. Lymphocytes were predominantly small-to-medium sized and had normal cytological features. There was also an increased population of large blast-like cells with prominent nucleoli and perinuclear clearing resembling lymphoblasts. Plasma cells were present in small numbers. The myeloid: erythroid ratio was within normal range (3.04) but the overall number of erythroid and myeloid cells was very low. In particular, very few erythroid cells were present. In the myeloid lineage there was also maturation disruption since immature ring form neutrophils (neutrophilic metamyelocytes) predominated over mature neutrophils (condensed chromatin) (*Figure 1D*). Giemsa staining of BM smear revealed a massively increased lymphoblast population and severe depletion of both immature erythroid cells and granulocytes in the cKO mice (*Figure 1E*). These cells were replaced by cells with prominent nucleoli and perinuclear clearing resembling lymphoblasts. To confirm that the lymphoblasts were the CD11b$^+$ Gr-1$^-$ cells identified

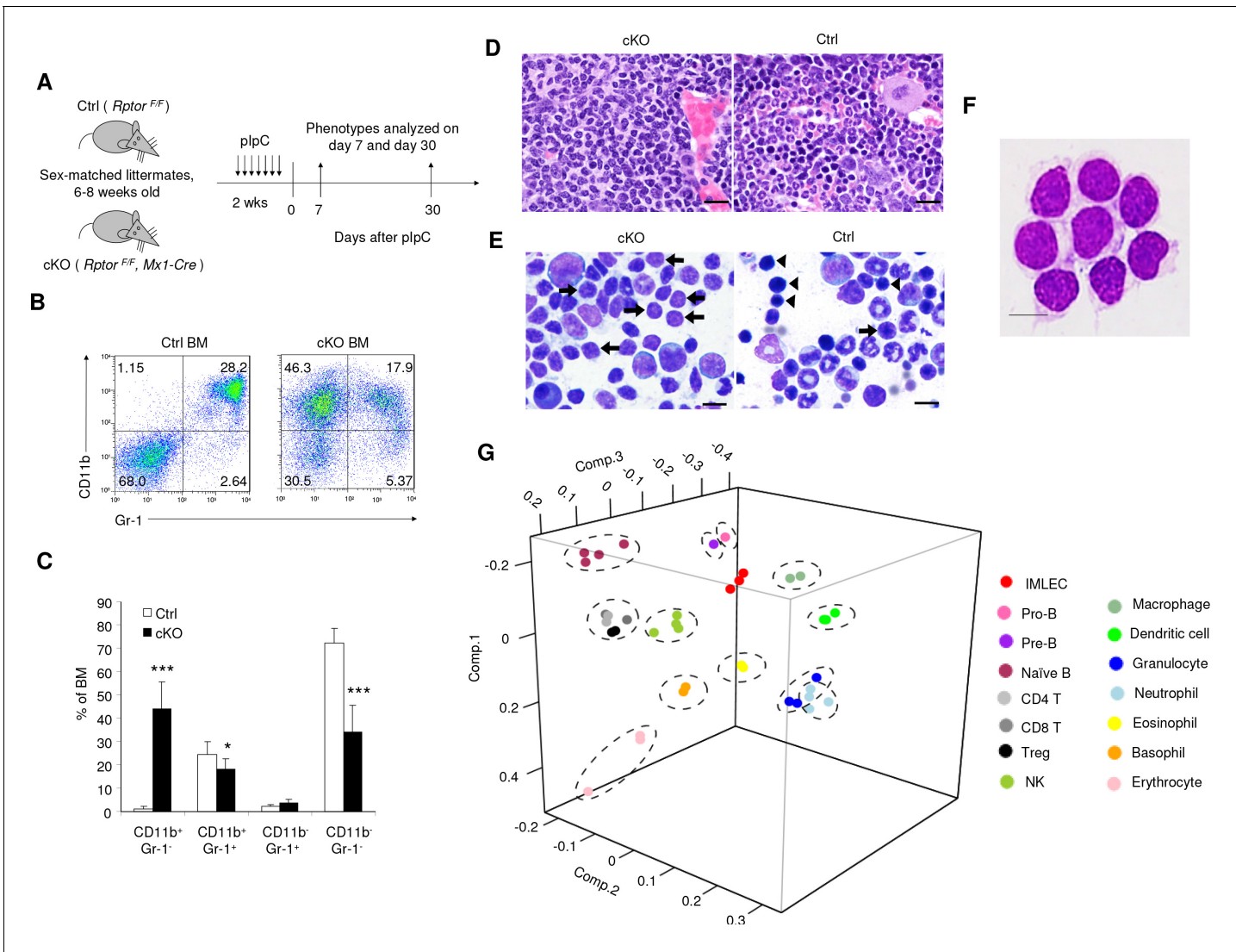

**Figure 1.** Targeted mutation of *Rptor* in hematopoiesis led to massive accumulation of IMLEC. (A) Schematic of experimental design. Sex-matched 6–8 weeks old Ctrl (*Rptor*^F/F^) and cKO (*Rptor*^F/F^, *Mx1-Cre*) mice were treated with pIpC for seven times. The phenotypes were analyzed on day 7 or day 30 after the complement of pIpC treatment. (B) Representative flow cytometric analysis of myeloid cells from mice BM by CD11b and Gr-1 on day seven after pIpC treatment. Similar data were obtained on day 30. (C) Frequencies of various hematopoietic cell populations in BM based on CD11b and Gr-1 markers. n = 7 for Ctrl mice; n = 9 for cKO mice. Data are pooled from three independent experiments. (D) Expansion of lymphoblast population in cKO BM revealed by histology. Left, H&E staining showing prominence of cells with lymphoid morphology in the cKO BM; Right, normal BM histology from Ctrl mouse showing normal myeloid and erythroid lineages. Scale bar, 20 μm. (E) Expansion of lymphoblast revealed by cytology. Left, cytological preparation of BM smear from cKO mouse showing preponderance of lymphocytes (short arrows) and severe depletion of myeloid and erythroid lineages. There were some promyelocytes and myelocytes present (larger cells) but no mature neutrophils. Right, normal cytological preparation of BM from Ctrl mouse. There were numerous erythroid precursors with intensely basophilic, condensed chromatin (arrowheads) and a few lymphocytes (short arrow) with less condensed chromatin. Additionally, in Ctrl BM there were numerous mature neutrophils (ringed nucleus with constrictions) which were severely depleted in the cKO mouse. Scale bar, 10 μm. (F) Giemsa staining of FACS-sorted *Rptor*-deficient CD11b^+^Gr-1^−^ BM cells. Scale bar, 10 μm. Similar morphology was observed in three independent experiments. (G) Principal component analysis (PCA) of gene expression in CD11b^+^ Gr-1^−^ BM cells (IMLECs) and other hematopoietic cells. Numbers along axes indicate relative scaling of the principal variables. RNA-seq data from IMLECs obtained in our study were compared with those deposited in public database by others. Datasets are from known lymphoid (Pro-B, Pre-B, Naïve B, CD4 T, CD8 T, Treg and NK) or myeloid (macrophage, dendritic cell, granulocyte, neutrophil, eosinophil, basophil and erythrocyte) subsets.

DOI: https://doi.org/10.7554/eLife.32497.003

The following figure supplements are available for figure 1:

**Figure supplement 1.** Conditional deletion of *Rptor* resulted in abnormal hematopoiesis.

DOI: https://doi.org/10.7554/eLife.32497.004

**Figure supplement 2.** Raptor deletion led to impaired developments of B lymphoid, erythroid and myeloid compartments in BM.

*Figure 1 continued on next page*

*Figure 1 continued*

DOI: https://doi.org/10.7554/eLife.32497.005

**Figure supplement 3.** *Rptor* cKO mice are pancytopenic.

DOI: https://doi.org/10.7554/eLife.32497.006

**Figure supplement 4.** *Rptor* deletion increased hematopoietic stem and progenitor cells in BM.

DOI: https://doi.org/10.7554/eLife.32497.007

**Figure supplement 5.** Tamoxifen induced conditional deletion of *Rptor* in hematopoiesis also led to massive accumulation of IMLECs.

DOI: https://doi.org/10.7554/eLife.32497.008

by flow cytometry in the cKO BM (*Figure 1B,C*), we FACS-sorted the subset based on CD11b$^+$ Gr-1$^-$ surface markers and validated its lymphoid morphology (*Figure 1F*).

The spleen was greatly enlarged due to expansion of the follicular centers and periarteriolar sheaths within the splenic white pulp (lymphoid areas) of the cKO mice (*Figure 1—figure supplement 1C*). The expanding white pulp populations consisted of lightly stained, large cells that morphologically resembled germinal center lymphocytes. In some areas, these populations expanded within the marginal zones while in others, they involved the periarteriolar sheaths. The cells had an increased amount of pale eosinophilic cytoplasm and mild pleomorphism with both centroblast-like cells (larger cells with large ovoid nuclei and 1–2 prominent nucleoli per cell) and centrocyte-like cells (smaller cells with cleaved or elongated nuclei and unapparent nucleoli).

Since cells with such combination of morphology and surface markers had not been identified previously, we FACS-sorted the CD11b$^+$ Gr-1$^-$ cells from the *Rptor* cKO BM and carried out next-generation RNA sequencing (RNA-seq). Using principal component analysis (PCA), we compared gene expression profiles of these CD11b$^+$ Gr-1$^-$ cKO BM cells with other known subsets of hematopoietic cells, including B cells, T cell subsets, NK cells, myeloid cell subsets, dendritic cells and erythroid cells. This analysis demonstrates that the CD11b$^+$Gr-1$^-$ cKO BM cells were distinct from all known blood cell types, although they appear to be closely related to B lymphocytes and macrophages (*Figure 1G*). We hereafter refer these cells as innate myeloidlymphoblastoid effector cells (IMLECs).

We identified subset specific genes using a threshold of 4-fold changes and an adjusted FDR-adjusted p-value<0.01. As shown in *Figure 2A*, by comparing RNA-seq data-based gene expression signature of all known leukocyte subsets, a unique gene expression signature was identified in the CD11b$^+$ Gr-1$^-$ cKO BM IMLECs. The signature consists of 48 genes that are up-regulated by more than 4-fold. The 48 genes over expressed in the CD11b$^+$ Gr-1$^-$ cKO BM IMLECs are listed in *Figure 2B*. Among their diverse functions, these genes are involved in intracellular signaling cascades (such as *Arhgap31*, *Rab20*, *Gna12*, *Mink1* and *Prkch*) and metabolic processes (such as *Naga*, *Atf3*, *Aoah*, *Chst14* and *Gns*). The uniqueness of cKO BM IMLEC is also supported by pair-wise comparisons between IMLEC and peritoneal macrophage or other closely related cell types that are prominent in BM (*Figure 2—figure supplement 1*).

A defining feature of the B cell lineage is activation of *Ig* gene loci, as evidenced by 'sterile' transcripts transcribed from the unarranged loci of Ig heavy chain (*Igh*) and light chains (*Igk* and *Igl*) (*Sleckman et al., 1996*). RNA-seq data revealed high levels of sterile transcripts within the *Igh* locus (*Figure 2C*) and *Igk* and *Igl* loci (*Figure 2—figure supplement 2*) from both B cell lineages and cKO BM IMLECs. This is a significant difference from macrophage, which had no detectable expression of the sterile transcripts, as expected. Another defining feature of developing B-lymphocytes is *Ig* gene rearrangement, a unique mechanism of genetic recombination that occurs only during the early stages of B cell maturation. This process is strictly dependent on recombinases genes *Rag1* (*Mombaerts et al., 1992*; *Schatz et al., 1989*) and *Rag2* (*Shinkai et al., 1992*; *Oettinger et al., 1990*). As shown in *Figure 2D*, no expression of *Rag1* was detectable by quantitative PCR (qPCR), although a detectable but extremely low level of *Rag2* was observed. Consistent with lack of *Rag1* expression, no gene rearrangement was found in the *Ig* loci (*Figure 2E*). Taken together, our data so far demonstrate that Raptor suppresses accumulation of a previously uncharacterized leukocyte subset with features of both myeloid and lymphoid cells.

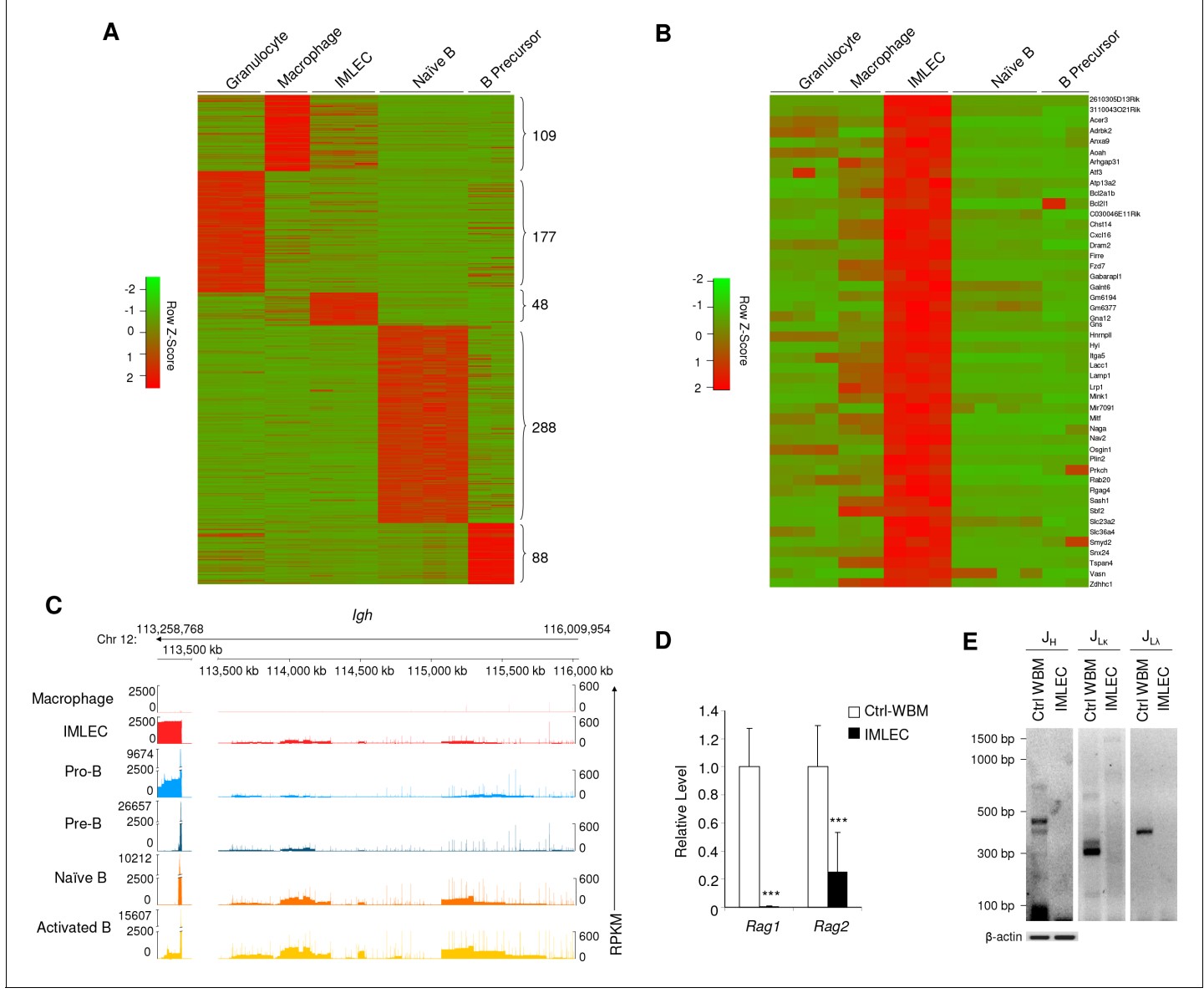

**Figure 2.** Unique gene signature of *Rptor*$^{-/-}$ CD11b$^+$ Gr-1$^-$ BM IMLECs. (**A**) Heat map representing the relative expression levels for indicated population-specific genes. Genes up-regulated for at least 4-fold with FDR-adjusted p-value<0.01 were considered population-specific. Numbers on right indicate amounts of population-specific genes. (**B**) Heat map representing expression levels of 48 CD11b$^+$Gr-1$^-$ BM IMLEC-specific genes among indicated populations including macrophage, B precursor, naive B and granulocyte. (**C**) Genome browser display of transcript structure and gene expression quantity for immunoglobulin heavy chain (IgH) complex gene region from indicated populations. For each track, the normalized numbers of aligned reads count or Reads Per Kilobase of transcript per Million mapped RNA (RPKM) are shown in y-axis, while the gene positions for the sterile transcripts are shown in x-axis. Numbers on top indicate the start and end loci on bases along chromosome 12 for specified gene region. Similar profiles of cKO IMLEC were observed in two other samples analyzed. (**D**) mRNA levels of *Rag1* and *Rag2* in cKO CD11b$^+$ Gr-1$^-$ BM IMLECs in comparison to Ctrl whole BM cells. n = 6 for Ctrl-WBM; n = 8 for IMLEC. Data are pooled from three independent experiments. (**E**) IMLECs did not display immunoglobulin gene rearrangement. FACS-sorted CD11b$^+$ Gr-1$^-$ cells (IMLECs) from cKO BM were dissected for Ig gene rearrangement patterns of heavy chain (J$_H$), light chains *Igk* (J$_{L\kappa}$) and *Igl* (J$_{L\lambda}$) by PCR. WBM cells from Raptor Ctrl mice were used as positive control. These data have been repeated twice.

DOI: https://doi.org/10.7554/eLife.32497.009

The following figure supplements are available for figure 2:

**Figure supplement 1.** In silico pair wise comparisons between IMLECs and other closely related leukocytes based on our RNA-seq data and the publically available RNA-seq data.

DOI: https://doi.org/10.7554/eLife.32497.010

**Figure supplement 2.** cKO IMLECs have a similar expression patterns as that of B-lymphoid subsets in the sterile transcripts from *Igk* and *Igl* loci.

*Figure 2 continued on next page*

*Figure 2 continued*

DOI: https://doi.org/10.7554/eLife.32497.011

## Characterization of IMLECs in *Rptor* cKO and WT mice

IMLECs identified in the *Rptor* cKO mice did not express surface markers that are used to define other lymphocytes, such as B220 for B cells, CD3 for T cells and NK1.1 for NK cells (*Figure 3A*). The high levels of CD11b indicated that these cells are distinct from the recently identified innate lymphoid cells (ILCs) that are CD11b negative (*Walker et al., 2013*). In addition, they lack surface markers for progenitor cells, such as c-Kit and Sca-1 (*Figure 3A*). Although the cKO IMLECs retained a high level of myeloid marker CD11b, they expressed a very low level of F4/80 macrophage marker and lacked CD115 monocyte marker (*Figure 3B*). To identify a positive marker for IMLEC, we searched our RNA-seq database for overexpression of genes that encode cell surface CD (cluster of differentiation) markers. Among 317 CD markers (*Supplementary file 1*), the most up-regulated gene in the cKO BM IMLECs over Ctrl whole BM cells was *Cd274* (also called *B7h1* or *Pdl1*). As shown in *Figure 3B* and *Figure 3C*, PD-L1 was expressed on the vast majority of CD11b$^+$ Gr-1$^-$ BM cells from *Rptor* cKO mice.

Based on the above data and availability of robust cell surface markers, we define IMLECs by their expression of CD11b and PD-L1, but lack of major lineage markers for T cells (CD3), B cells (B220), natural killer cells (NK1.1), erythroid cells (Ter119), granulocytes (Gr-1), macrophages and monocytes (F4/80 and CD115). These markers allowed us to search wild-type BM for IMLEC. Interestingly, a clear although small fraction of the Lin$^-$ (CD3$^-$B220$^-$ NK1.1$^-$ Ter119$^-$ Gr-1$^-$ F4/80$^-$ CD115$^-$) CD11b$^+$ BM cells in the Ctrl mice also expressed PD-L1 (*Figure 3C*, left panel), although the overall PD-L1 expression level was not as high as that from cKO IMLECs. Following *Rptor* deletion, a robust expansion (approximately 500-fold) of Lin$^-$ CD11b$^+$ PD-L1$^+$ BM IMLECs was observed (*Figure 3C*, right panel).

It should be noted that although cKO IMLECs also over-express *Cd11c* gene (*Supplementary file 1*), IMLEC gene expression profiles are distinct from dendritic cell (DC) based on gene signature (*Figure 1G* and *Figure 2—figure supplement 1E,F*). In cKO IMLECs, the CD11c levels were somewhat lower than the PD-L1$^-$ DC (*Figure 3D*). In WT mice, greater than 90% of Lin$^-$CD11b$^+$PD-L1$^+$ IMLEC in BM, lung and peripheral blood mononuclear cells expressed only low levels of CD11c, while those in spleen and mesenteric lymph nodes consisted of two major subsets: CD11c$^{high}$ and CD11c$^{low/-}$ (*Figure 3E*). IMLECs were also found among the leukocytes isolated from lung and in peripheral lymphoid organ (*Figure 3F, G and H*), and this population was greatly expanded in the cKO mice (*Figure 3I*).

To further confirm that this subset is the IMLEC in normal BM, we FACS-sorted the Lin$^-$ (CD3$^-$ B220$^-$ NK1.1$^-$ Ter119$^-$ Gr-1$^-$ F4/80$^-$ CD115$^-$) CD11b$^+$ PD-L1$^+$ cells from wild type (WT) BM and characterized their morphology and levels of sterile *Ig* transcripts. As shown in *Figure 4A*, the sorted cells had a lymphoid morphology as did the cKO IMLECs. They also displayed comparable size and granularity as cKO IMLECs (*Figure 4B*). Moreover, Lin$^-$ CD11b$^+$ PD-L1$^+$ cells from WT BM expressed sterile transcripts of *Ig* loci identified by RNA-seq (*Figure 4C*). Furthermore, subsequent validation of IMLECs in WT BM was undertaken by comparing the expression of other top candidate markers (CD14, CD16) and MHC-I/MHC-II (*Figure 4—figure supplement 1A*), as well as population-specific transcription factors *Mitf*, *Atf3* and *Zdhhc1* (*Figure 4D,E*). The largely comparable expression levels of these surface markers and transcription factors between WT and cKO IMLECs provide additional lines of evidence for these cells to be naturally occurring IMLECs. Therefore, a small fraction of normal leukocytes in lymphoid and non-lymphoid tissues have the IMLEC phenotype, and this subset is massively expanded after *Rptor* deletion.

## Altered differentiation is responsible for accumulation of IMLECs

Theoretically, expansion of IMLECs in cKO BM may be caused by increased proliferation and/or reduced apoptosis. To test this possibility, we analyzed the proliferation of IMLECs by Ki-67 staining and BrdU incorporation. Remarkably, Lin$^-$ CD11b$^+$ PD-L1$^+$ IMLECs from both Raptor Ctrl and cKO BM had much fewer Ki-67$^+$ cells (*Figure 4F,G*) or BrdU$^+$ cells (*Figure 4—figure supplement 1B,C*) when compared with other lineages. The fact that IMLECs are not proliferating at a higher rate than

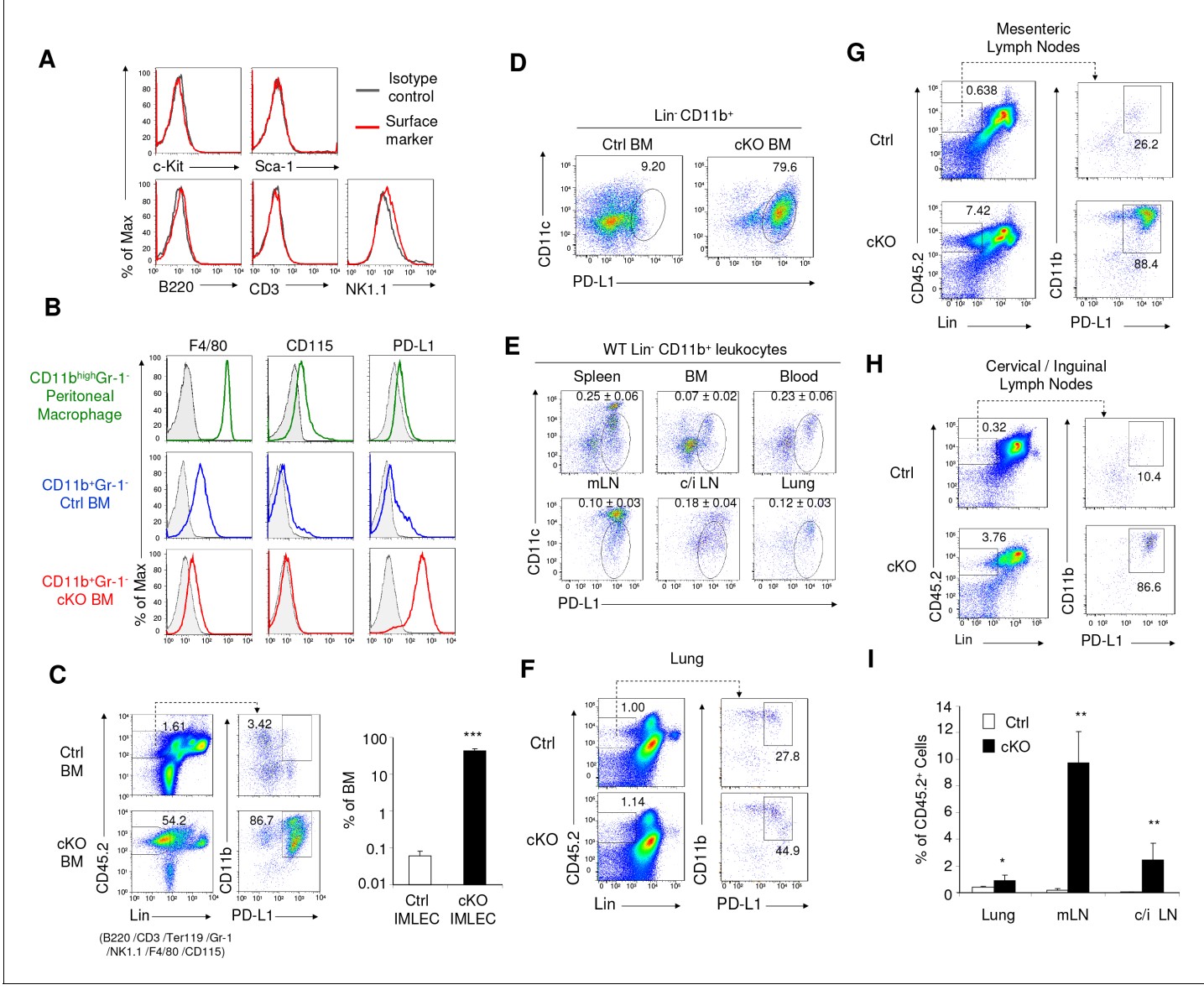

**Figure 3.** Identification of IMLEC by cell surface markers in both Raptor cKO and WT mice. (A) IMLECs do not express surface markers for conventional lymphocytes (CD3, B220 and NK1.1) and stem and progenitor cells (c-Kit and Sca-1). CD11b$^+$ Gr-1$^-$ IMLECs from Raptor cKO BM were tested for their expression of lymphocytes and progenitor cell markers. These data have been repeated three times. (B) IMLECs do not express markers for monocytes and macrophages but surprisingly express high levels of PD-L1. Filled gray areas indicate distributions of fluorescence from stainings by control antibodies. One representative result of at least five experiments is shown. (C) *Rptor* deletion causes expansion of Lin$^-$ (B220$^-$ CD3$^-$ Ter119$^-$ NK1.1$^-$ Gr-1$^-$ F4/80$^-$ CD115$^-$) CD11b$^+$ PD-L1$^+$ IMLECs in BM. Representative flow staining profiles for BM IMLECs (left) and their abundance summary data (right) are presented. n = 5 for Ctrl BM; n = 5 for cKO BM. (D) Lin$^-$CD11b$^+$ PD-L1$^+$ BM IMLECs are CD11c$^{-/low}$. One representative result of two experiments is shown. (E) The Lin$^-$ CD11b$^+$ PD-L1$^+$ CD11c$^{-/low}$ IMLECs are found in various lymphoid and non-lymphoid organs from WT mice. mLN, mesenteric lymph nodes; c/i LN, cervical and inguinal lymph nodes. The numbers (Mean ±SD) are summarized abundances of IMLECs among mononuclear cells (MNCs) from 3 WT mice. (F–H) *Rptor* deletion caused a broad accumulation of IMLECs in lung (F), mesenteric lymph nodes (G) and peripheral (cervical and inguinal) lymph nodes (H). Results shown are representative of three mice in each group. (I) Summary data showing increased IMLECs in the lymphoid and non-lymphoid organs. n = 3 for both groups.
DOI: https://doi.org/10.7554/eLife.32497.012

other BM cell types effectively rules out rapid proliferation as an explanation for IMLECs accumulation in Raptor cKO BM. Likewise, the massive increase of IMLEC in cKO mice over those in the Ctrl mice cannot be due to proliferation, as the percentage of Ki-67$^+$ or BrdU$^+$ cells is not increased in cKO mice. Furthermore, based on cell surface Annexin V staining, IMLECs from cKO BM were more

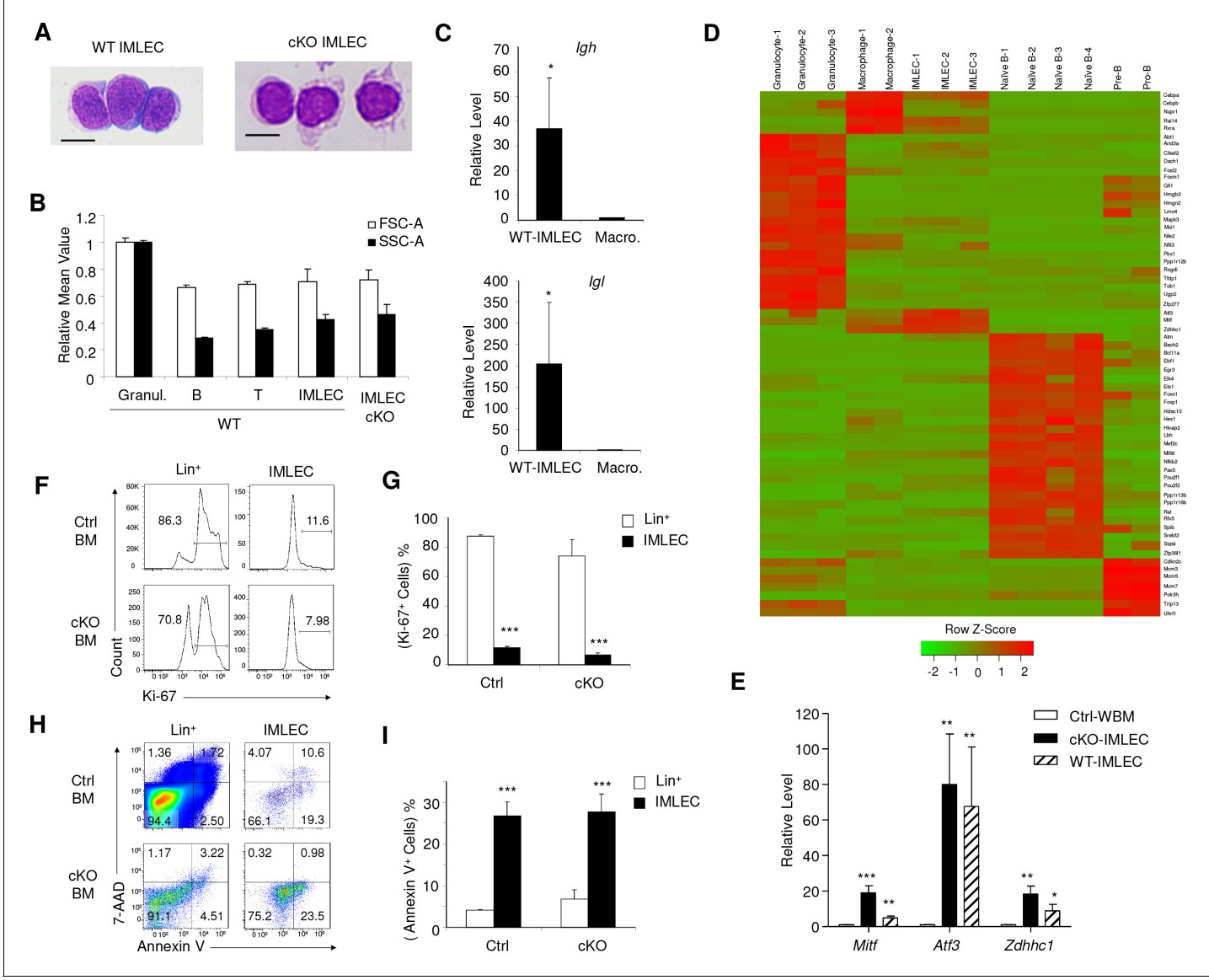

**Figure 4.** Comparison of IMLECs identified in both WT and Raptor-deficient BM. (**A**) Giemsa staining of Lin⁻ CD11b⁺ PD-L1⁺ IMLECs from WT BM and *Rptor*-cKO BM. One representative result of 3 independent experiments is shown. Scale bar, 10 μm. (**B**) Size and granularity of IMLECs from Raptor Ctrl and cKo BM were evaluated by FSC and SSC, respectively. n = 5 for each group. (**C**) IMLEC from WT BM expresses sterile transcripts from the *Igh* and *Igl* loci. *Ig* transcript levels in macrophage are artificially defined as 1. (**D**) Heat map representing the relative expression levels for indicated population specific transcription factor genes. Genes up-regulated for at least 4-fold with FDR-adjusted p-value<0.01 were considered population-specific. (**E**) IMLECs from both WT and Raptor-cKO BM exhibit high levels of mRNA transcripts for the specified transcription factors (*Mitf*, *Atf3* and *Zdhhc1*). n = 5 for each group. (**F, G**) Ctrl and cKO IMLECs are non-cycling cells in contrast to Lin⁺ BM cells. (**H, I**) IMLECs are prone to apoptosis based on their binding to Annexin V. Results in (**C**), (**G**) and (**I**) represent one of 2 independent experiments with each involving three mice per group.

DOI: https://doi.org/10.7554/eLife.32497.013

The following figure supplement is available for figure 4:

**Figure supplement 1.** Comparisons of surface markers, viability and cellular proliferation among IMLECs and other defined lineages.
DOI: https://doi.org/10.7554/eLife.32497.014

prone to apoptosis than total lineage⁺ population (*Figure 4H,I*) and had apoptosis rate that was comparable to granulocytes, B cells and T cells in BM (*Figure 4—figure supplement 1D,E*). The pronounced apoptosis also rules out the possibility that increased survival may account for preferential accumulation of IMLECs in cKO mice. The robust apoptosis detected among WT IMLECs likely contributed to the reduced amount of IMLECs in normal BM (*Figure 3C*). Consistent with the reduced

proliferation and increased apoptosis of IMLECs, our exhaustive efforts to demonstrate self-renewal of IMLEC through transplantation of massive numbers of IMLEC have all been unsuccessful (data not shown).

As an alternative hypothesis, we evaluated whether IMLECs accumulated because of altered differentiation of hematopoietic stem and progenitors (HSPCs). As the first step to test this hypothesis, we evaluated if IMLEC accumulation in cKO BM was cell-intrinsic. Briefly, we mixed either $Rptor^{F/F}$ or $Rptor^{F/F}$, $Mx1$-$Cre$ (both CD45.2$^+$) BM cells with recipient type CD45.1$^+$ WT BM cells at a 2:1 ratio. At six weeks after BM transplantation, $Rptor$ was deleted from the $Rptor^{F/F}$, $Mx1$-$Cre$ donor-derived cells by pIpC treatment (**Figure 5A**). As shown in **Figure 5B**, the accumulation of CD11b$^+$ Gr-1$^-$ IMLECs was intrinsic to $Rptor^{-/-}$ BM cells. Since our earlier data suggested that IMLECs accumulated at the expense of granulocytes (**Figure 1B,C**), we tested if granulocytes were converted to IMLECs following $Rptor$ deletion. We produced $Rptor^{F/F}$, $Lyz2$-$Cre^{+/+}$ mice that should have myeloid lineage-specific deletion of $Rptor$. However, despite the effective deletion of the $Rptor$ gene in the granulocytes (**Figure 5C**), the percentages of CD11b$^+$ Gr-1$^+$ granulocytes and CD11b$^+$ Gr-1$^-$ IMLECs were unchanged (**Figure 5D**). These data suggest that accumulation of IMLEC in the cKO mice was not due to trans-differentiation from granulocytes.

Next, we use both in vitro co-culture and in vivo BM transplantation to identify the progenitor that may give rise to IMLECs. We co-cultured OP9 stromal cells with FACS-sorted BM LSK (Lin$^-$ Sca-1$^+$ c-Kit$^+$), CMP (Lin$^-$ Sca-1$^-$ c-Kit$^+$ CD34$^{Medium}$ CD16/32$^{Medium}$) and CLP (Lin$^-$CD127$^+$ Sca-1$^{Medium}$c-Kit$^{Medium}$) populations from Ctrl and cKO mice that had been treated with pIpC (**Figure 5E**). As shown in **Figure 5F**, both LSK and CMP populations from $Rptor^{-/-}$ BM gave rise to CD11b$^+$ Gr-1$^-$ PD-L1$^+$ IMLECs. As expected, $Rptor$-sufficient CLPs were not able to give rise to CD11b$^+$ myeloid cells. Interestingly, $Rptor$-deficient CLPs generated progenies with a small portion exhibiting immunophenotypes of IMLECs. We also transplanted sorted LSK and CMP populations and induced $Rptor$ deletion in the donor cells by treating recipients with pIpC (**Figure 5G**) to confirm their ability in giving rise to IMLECs. Due to lack of self-renewal activity of progenitor cells and rapid apoptosis of IMLEC, we used a much shorter timeline than the whole bone marrow transplantation studies in order to capture progenitor-derived IMLEC. As shown in **Figure 5H**, deletion of $Rptor$ in either LSKs or CMPs was sufficient to induce the generation of CD11b$^+$ Gr-1$^-$PD-L1$^+$ cells in recipients BM. The shorter timeline explained relative paucity of LSK-derived IMLEC when compared with long-term bone marrow transplantation (**Figure 5A,B**). As expected, since CMPs do not have self-renewal capability, only a small number of progeny cells were produced. However, since IMLEC can be generated from CLP in vitro, their potential to do so under physiological conditions cannot be ruled out. Taken together, our data demonstrate that the massive accumulation of IMLECs in cKO mice can be caused by altered differentiation of CMPs, although other differentiation pathway cannot be ruled out.

## Reduced c-Myb expression is responsible for accumulation of IMLECs

A previous study demonstrated that heterozygous $Myb$ mutation leads to an expansion of BM CD11b$^+$Gr-1$^-$ cells (**García et al., 2009**). Although expression of PD-L1 was not evaluated in the earlier study, we were intrigued by the possibility that down-regulation of $Myb$ may be the underlying mechanism for the massive production of IMLECs. Since both LSKs and CMPs are able to give rise to CD11b$^+$ Gr-1$^-$ PD-L1$^+$ IMLECs, we evaluated expression of c-Myb in both LSK and CMP populations. Indeed, the $Myb$ transcripts were significantly reduced in both LSK and CMP populations sorted from $Rptor$ cKO mice (**Figure 6A**). Moreover, our intracellular staining also revealed reduced levels of c-Myb protein in both LSKs and CMPs from cKO BM (**Figure 6B**). Interestingly, induced deletion of c-Myb in mice with homozygous floxed c-Myb ($Myb^{F/F}$, $Mx1$-$Cre$), but not heterozygous floxed c-Myb ($Myb^{F/+}$, $Mx1$-$Cre$), showed obvious increase of PD-L1 expression in CD11b$^+$Gr-1$^-$ BM cells (**Figure 6C,D,E** and **Figure 6—figure supplement 1A,B,C**), despite of significant decrease of BM whole leukocytes due to $Myb$-reduction induced cell apoptosis. To avoid excessive apoptosis and test if the down-regulation of $Myb$ was necessary and sufficient to cause accumulation of IMLECs, we transplanted BM cells from either $Myb^{F/F}$, $Rag2^{-/-}$ or $Myb^{F/F}$, $Rag2^{-/-}$, $CreER$ mice to CD45.1$^+$ recipients, which then received tamoxifen to achieve deletion of $Myb$ specifically in donor-derived hematopoietic cells after their full reconstitution (**Figure 6F**). Consistent with essential role for c-Myb in hematopoiesis, surviving leukocytes appeared heterozygous for $Myb$ deletion (**Figure 6G**). As early as 1 week after the first injection of tamoxifen, a significant decrease in CD11b$^+$ Gr-1$^+$

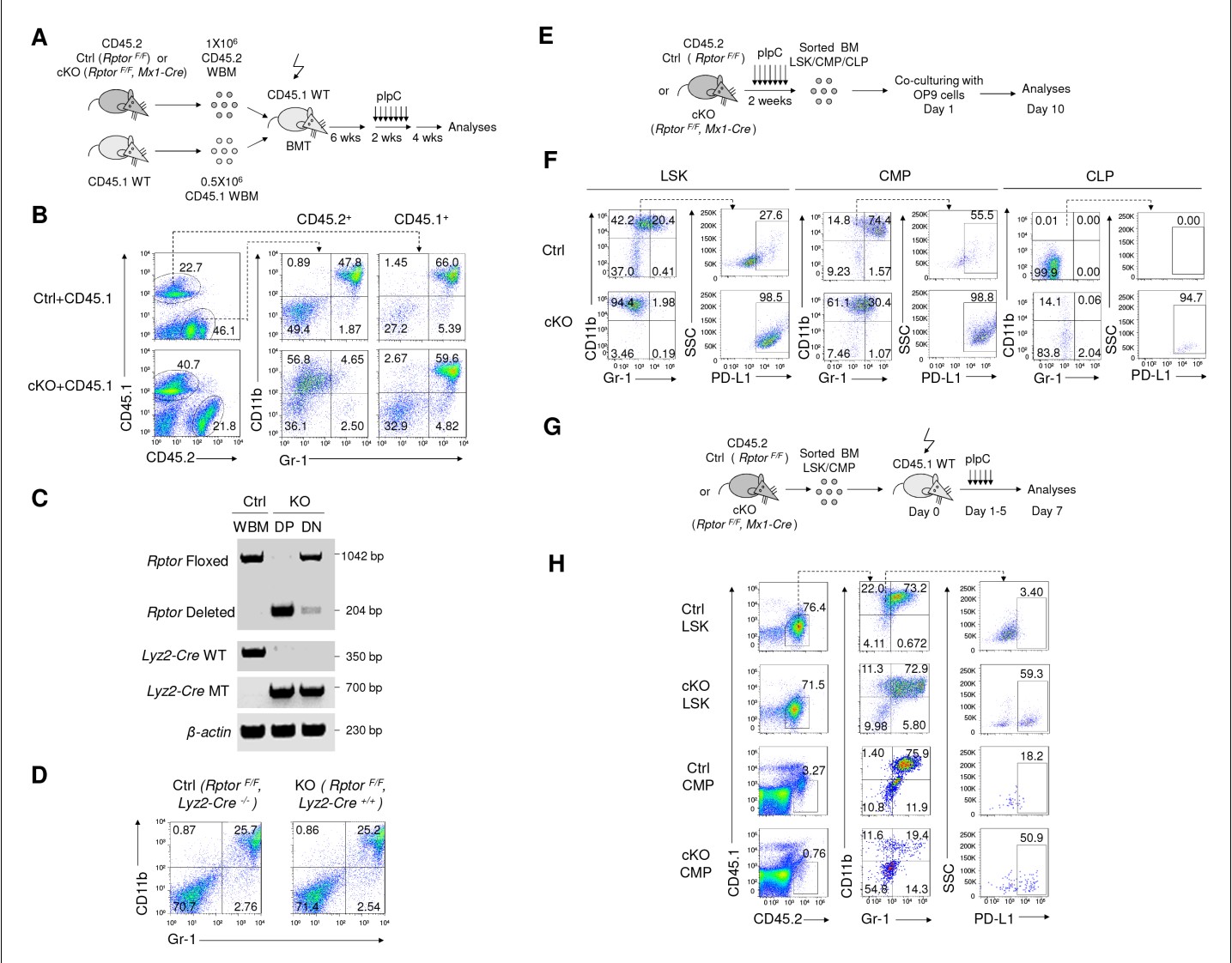

**Figure 5.** Raptor represses differentiation of IMLEC from CMP. (**A, B**) IMLEC accumulation in *Rptor*[-/-] BM is cell autonomous. (**A**) Diagram depicting generation of mixed BM chimera, induction of *Rptor* deletion and analyses for IMLEC. (**B**) In mixed BM chimera mice, IMLEC was accumulated only in the Raptor-deficient hematopoietic cells. Mice were sacrificed and analyzed at 4 weeks post pIpC treatment. CD45.1[+] and CD45.2[+] donor-derived BM cells were gated to show the myeloid subsets based on CD11b and Gr-1. Similar data were obtained in three independent experiments, each involving at least three mice per group. (**C, D**) BM IMLECs are not converted from CD11b[+] Gr-1[-] granulocytes. (**C**) Genotyping and deletion efficacy. CD11b[+]Gr-1[+] cells (DP) and CD11b[-]Gr-1[-] cells (DN) were sorted from BM of KO (*Rptor*[F/F], *Lyz2-Cre*[+/+]) mice. Floxed and deleted *Rptor* alleles, as well as *Lyz2-Cre* wild type (WT) and mutated (MT) alleles were confirmed by PCR. (**D**) Efficient deletion of *Rptor* in the CD11b[+]Gr-1[+] compartment failed to cause IMLEC accumulation. Data shown are representative BM flow profiles of Ctrl mice and those with *Rptor* deletion in the CD11b[+]Gr-1[+] compartment, depicting distributions of CD11b and Gr-1 markers in adult mice. Similar data have been obtained in three experiments, involving a total of 5 mice per group. (**E, F**) *Rptor*[-/-] LSK, CLP and CMP cells differentiate into IMLECs in vitro. (**E**) Diagram of experimental design. 2 × 10[3] LSK cells or 5 × 10[4] CLP or 5 × 10[4] CMP cells were co-cultured with the OP9 cells for 10 days. (**F**) *Rptor* deletion promoted generation of IMLEC. The CD45.2[+] leukocytes were gated and analyzed for their expression of CD11b, Gr-1 and PD-L1. Data shown are representative of three independent experiments. (**G, H**) Both LSK and CMP cells differentiate into IMLECs in vivo. (**G**) Diagram of BM cells transplantation. FACS-sorted, Ctrl or cKO LSK cells (5 × 10[4]/mouse) or CMP cells (1.2 × 10[5]/mouse) were injected i.v. to CD45.1 recipient mice which were immediately administrated with five daily pIpC treatments. BM cells were harvested on day seven for FACS analyses. (**H**) *Rptor* deletion promoted differentiation of progenitor cells into IMLEC. Donor-derived CD45.2[+] BM cells were gated to analyze surface markers CD11b, Gr-1 and PD-L1. Data shown are representative flow profiles from one of 3 independent experiments.
DOI: https://doi.org/10.7554/eLife.32497.015

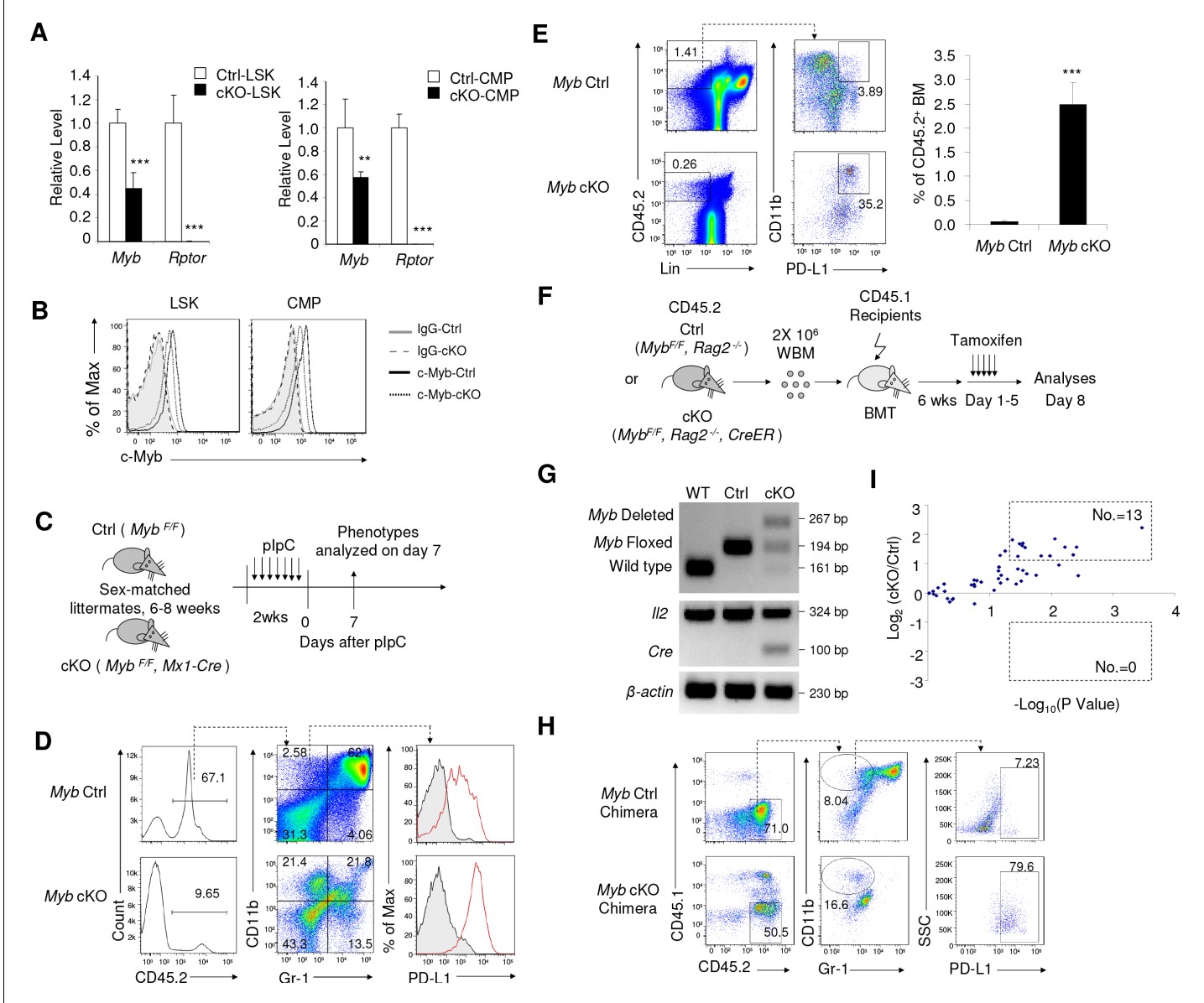

**Figure 6.** Inactivation of *Myb* is an underlying cause for accumulation of IMLEC. (**A**) Reduction of *Myb* mRNA in the *Rptor*-deficient HSPCs. BM LSK and CMP populations were FACS-sorted from Ctrl and cKO mice at 1–3 weeks after pIpC treatment. Quantitation of *Myb* mRNA was performed by qPCR. n = 5 for Ctrl-LSK; n = 4 for cKO-LSK (left). n = 4 for Ctrl-CMP; n = 3 for cKO-CMP (right). (**B**) Detection of c-Myb protein by intracellular staining. Data represent one of three independent experiments with similar results. (**C–E**) Deletion of c-Myb in mice with homozygous floxed *Myb* resulted in enhanced generation of IMLECs. (**C**) Schematic of experimental design. Sex-matched 6–8 weeks old c-Myb Ctrl (*Myb $^{F/F}$*) and cKO (*Myb $^{F/F}$, Mx1-Cre*) mice were treated with pIpC for seven times. The phenotypes were analyzed on day seven after the complement of pIpC treatment. Inducible deletion of c-Myb showed obvious increase of PD-L1 expression on CD11b$^+$Gr-1$^-$ BM cells (**D**) and production of IMLECs (**E**). n = 3 for c-Myb Ctrl mice; n = 3 for c-Myb cKO mice. (**F–H**) Deletion of *Myb* enhances generation of IMLEC. (**F**) Diagram of experimental design. Whole BM cells (2 × 10$^6$/mice) of given genotypes were used for transplantation. Once the chimera mice were established, deletion of *Myb* was induced by five daily injection of tamoxifen. (**G**) Detection of *Myb* deletion in the whole BM cells after tamoxifen treatments. Data are representative of two independent experiments. (**H**) Generation of IMLEC is promoted by inactivation of *Myb*. BM cells were harvested at 7 days after first tamoxifen treatment and analyzed for IMLECs based on surface markers CD11b, Gr-1 and PD-L1 within the donor-derived CD45.2$^+$ BM cells. Data shown represent one of three experiments using either first or second generation of BM chimeras. (**I**) *Rptor* deletion broadly increases miRNAs targeting *Myb*. Lin$^-$ c-Kit$^+$ HSPCs were isolated from Raptor Ctrl and cKO mice at 10 days after pIpC treatment. miRNA levels were measured by miRNA microarray. The y-axis shows the log$_2$ ratio of signal, while the x-axis shows the -log$_{10}$P value. The doted boxes show the numbers of significantly (p<0.05) up-regulated (fold change >2) or down-regulated (fold change <0.5) miRNAs among 50 miRNAs with mirSVR score <−1.0. Each dot represents the mean value of a unique miRNA from three independent samples.

*Figure 6 continued on next page*

Figure 6 continued

DOI: https://doi.org/10.7554/eLife.32497.016

The following figure supplement is available for figure 6:

**Figure supplement 1c.** Myb expression and accumulation of IMLECs.

DOI: https://doi.org/10.7554/eLife.32497.017

granulocytes and an increase in CD11b$^+$ Gr-1$^-$ PD-L1$^+$ IMLECs were observed in *Myb* cKO BM (*Figure 6H*). Moreover, the provision of heterologous c-Myb significantly diminished the generation of IMLECs from Raptor-deficient LSK cells in our in vitro OP9 co-culturing experiments (*Figure 6— figure supplement 1D,E,F,G*). Therefore, down-regulation of *Myb* is necessary for production of IMLECs.

We recently reported that deletion of *Rptor* caused up-regulation of miRNA biogenesis in HSPCs (*Ye et al., 2015*). We searched our miRNA microarray database and mirSVR score database (*Betel et al., 2010*) for potential impact of miRNAs in down-regulation of *Myb* expression. Using the stringent criteria of mirSVR score <−1.0, we identified 50 miRNAs that presented in the HSPCs (*Supplementary file 2*). Among them, 13 miRNAs showed >2.0 folds up-regulation in the *Rptor*-deficient HSPCs (p<0.05), while none showed statistically significant down-regulation (*Figure 6I*). Significant up-regulation of miR-150 (1.4 folds increase, p=0.05), which was previously demonstrated to inhibit *Myb* expression (*Lin et al., 2008*; *Xiao et al., 2007*), was also observed. Up-regulation of *Myb*-targeting miRNAs provides a plausible mechanism for down-regulation of *Myb* by *Rptor* deletion. However, the broad spectrum of the up-regulated miRNAs suggests that it is unlikely that a single miRNA is responsible for the overall reduction of *Myb* expression.

## Expansion of IMLECs associated with lethal inflammatory response to TLR ligands

RNA-seq data indicated that IMLECs broadly up-regulate pattern recognition receptors (PRRs) genes. TLRs, the first family of PRRs identified, were broadly over-expressed in cKO IMLECs as determined by RNA-seq (*Figure 7A*). RT-PCR confirmed that IMLECs from both cKO and WT mice over-expressed essentially all TLRs tested, particularly *Tlr2*, *Tlr3*, *Tlr4*, *Tlr6*, *Tlr7*, *Tlr8* and *Tlr9* (*Figure 7B*). In addition to TLRs, expression of other PRRs, including NLRs, ALRs and RLRs, was also broadly elevated (*Figure 7—figure supplement 1*). When the BM from Ctrl and *Rptor*$^{-/-}$ mice were compared for their responses to TLRs agonists, including synthetic *tripalmitoylated lipopeptide* Pam3CysSerLys4 (Pam3CSK4, TLR1/2 agonist), heat-killed *Listeria monocytogenes* (HKLM, TLR2 agonist), synthetic analog of double-stranded RNA poly I:C (pIpC, TLR3 agonist), lipopolysaccharide (LPS, TLR4 agonist), flagellin from *Salmonella typhimurium* (FLA-ST, TLR5 agonist), synthetic lipoprotein derived from *Mycoplasma salivarium* (FSL-1, TLR2/6 agonist), single-stranded RNA Double-Right complexed with LyoVec (ssRNA-DR, TLR7 agonist) and synthetic oligonucleotides containing unmethylated CpG dinucleotides (ODN1826, TLR9 agonist), it is clear that *Rptor*$^{-/-}$ BM increased production of TNF-$\alpha$ (*Figure 7C*) and MCP-1 (*Figure 7D*) in response to all TLR ligands tested. To determine the cellular basis for the enhanced cytokine production, we used intracellular and cell surface staining to identify cells that produced the inflammatory cytokines. As shown in *Figure 7E*, IMLECs from both WT and *Rptor* cKO BM were potent producers of TNF-$\alpha$ when stimulated by LPS.

Consistent with exacerbated responses to TLR ligands in BM cells, deletion of *Rptor* in cKO mice resulted in massive increase in inflammatory cytokines in serum (*Figure 8A*). Approximately 40% of cKO mice died within 2 months after 7 pIpC treatments (*Figure 8B*). Histological analyses revealed extensive inflammation in the liver with associated tissue injuries (*Figure 8C*). A substantial proportion of the leukocytes in the liver of cKO mice were IMLECs, as demonstrated by cell surface markers CD11b$^+$ Gr-1$^-$ PD-L1$^+$ F4/80$^{low/-}$ (*Figure 8D,E*). To test if the mice with expanded IMLECs were more sensitive to endotoxin, we challenged the Ctrl and cKO mice with low doses of LPS (5 mg/kg body weight). While all Ctrl mice survived the LPS challenge, all cKO mice succumbed within 36 hr (*Figure 8F*). The dramatically increased mortality due to endotoxic shock was associated with remarkably elevated levels of inflammatory cytokines. As shown in *Figure 8G* and *Figure 8H*, a more than 500-fold increase in TNF-$\alpha$ and an approximately 10-fold increase of MCP-1 were detected in the serum of cKO mice at 6 hr after LPS injection. It has been demonstrated that

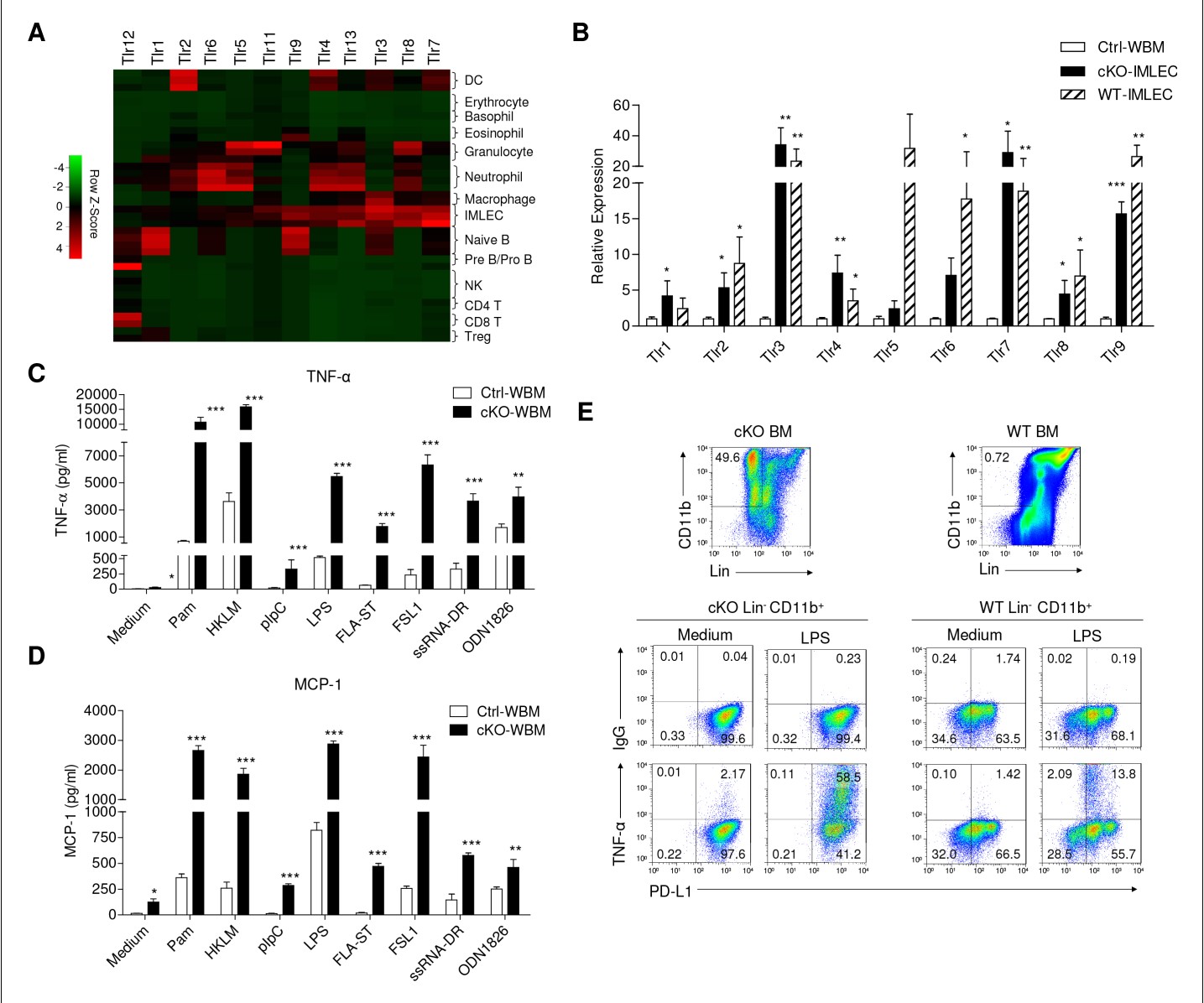

**Figure 7.** IMLECs broadly express PPRs and produce large amounts of inflammatory cytokines upon stimulation by various TLR ligands. (**A**) Heat map showing the relative transcript levels of TLRs among the indicated populations. (**B**) Both WT and Raptor-cKO IMLECs have greatly elevated expression of multiple TLRs genes in comparison to Ctrl BM. q-PCR was performed to determine transcript levels of *Tlr1-9* genes. After normalizing for cDNA input based on *Hprt* mRNA in each sample, the *Tlr1-9* levels in the FACS-sorted cKO CD11b$^+$ Gr-1$^-$ BM IMLEC were compared with Ctrl BM (artificially defined as 1.0). n = 3 for Ctrl WBM; n = 5 for cKO- IMLEC; n = 5 for WT-IMLEC. Similar results were obtained using mice sacrificed at 2 weeks ~2 months after pIpC treatment. (**C, D**) In responses to various TLR ligands, *Rptor* cKO BM cells produced greatly elevated amounts of TNF-α (**C**) and MCP-1 (**D**) than the Ctrl BM. Data shown are from one experiment involving three repeats per group and have been reproduced in five independent experiments. (**E**) Lin$^-$ (B220$^-$CD3$^-$Ter119$^-$Gr-1$^-$NK1.1$^-$F4/80$^-$CD115$^-$) CD11b$^+$ PD-L1$^+$ IMLECs from both cKO and WT BM were robust TNF-α producers after stimulation with LPS. BM cells (1 × 10$^7$ cells / well) were stimulated with LPS (1 μg/ml) for 16 hr with the presence of Golgi blocker in the last 4 hr. Data shown are representative profiles from one experiment and have been reproduced in three independent experiments.

DOI: https://doi.org/10.7554/eLife.32497.018

The following figure supplement is available for figure 7:

**Figure supplement 1.** IMLECs broadly over-express multiple families of pattern recognition receptors (PRRs) when compared with other blood cells.
DOI: https://doi.org/10.7554/eLife.32497.019

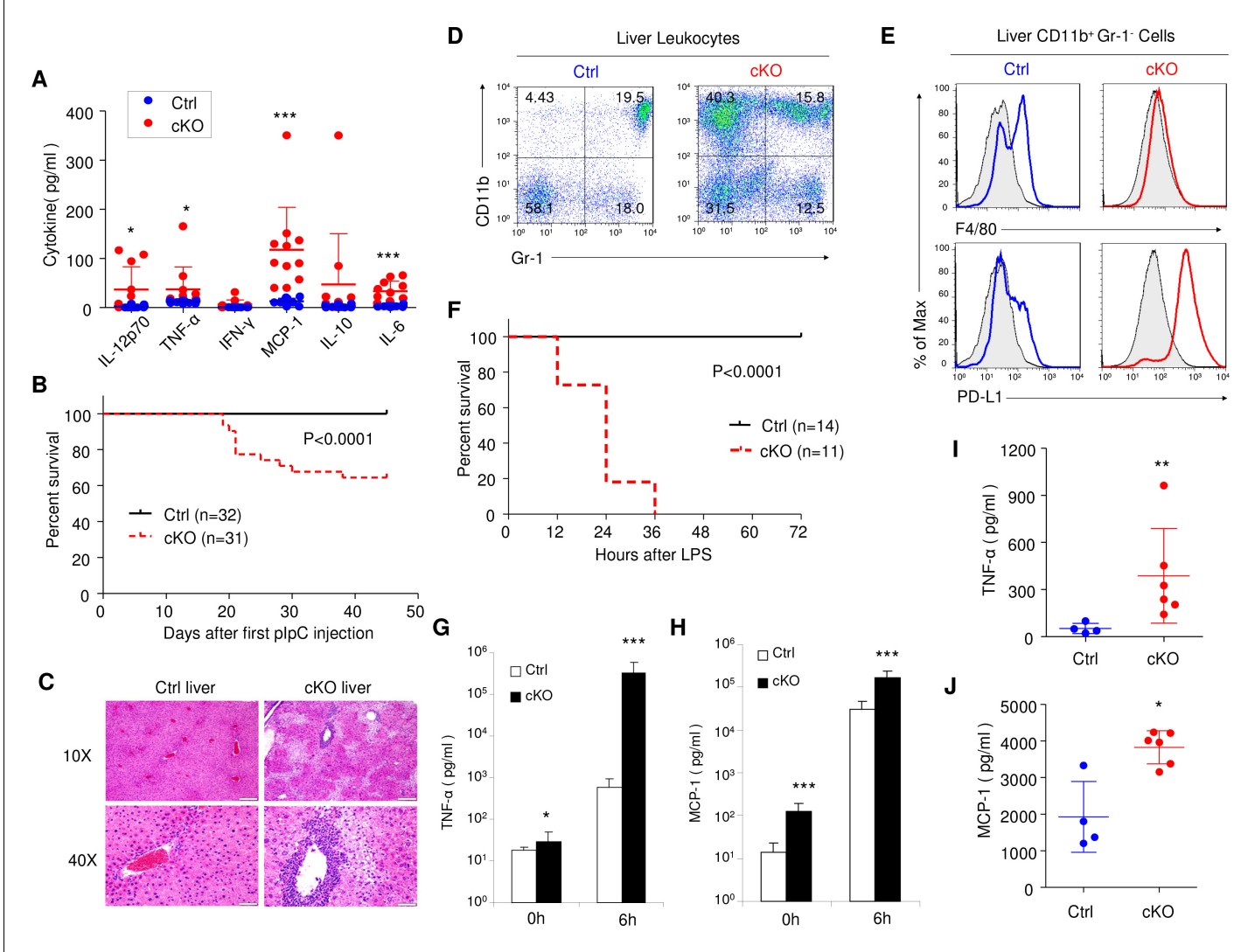

**Figure 8.** Rptor cKO mice are hyper sensitive to challenges by TLR ligands. (**A**) High blood levels of inflammatory cytokines in cKO mice that have received 7 pIpC treatments. Serum samples collected at 7 days after the last pIpC treatment were tested for levels of IL-12p70, TNF-α, IFN-γ, MCP-1, IL-10 and IL-6. Dot plots depict cytokine levels in individual cKO (red dots) or Ctrl (blue dots) mice, and line drawings depict means and SD. n = 12 for Ctrl mice and n = 11 for cKO mice. (**B**) High proportions of *Rptor* cKO mice that received 7 pIpC injections died within 7 weeks after first pIpC treatment. Data shown are Kaplan-Meier survival curve, and the statistical significance is determined by log-rank test. n = 32 for Ctrl mice and n = 31 for cKO mice. (**C**) Representative H and E staining of liver sections from *Rptor* Ctrl and cKO mice at 2 weeks post pIpC treatment. Scale bars represent 200 μm (10X) and 50 μm (40X), respectively. (**D, E**) Massive accumulation of IMLEC in cKO liver. (**D**) Accumulation of CD11b+ Gr-1- cells in the cKO liver. (**E**) CD11b+ Gr-1- cells in cKO livers demonstrated IMLEC surface markers, including PD-L1high and F4/80low/-. Gray lines indicate the staining for isotype control antibodies. Results are representative of at least five independent analyses. (**F–H**) cKO mice are vulnerable to low doses of LPS challenges. At 2–3 months after pIpC treatments, cKO and Ctrl mice were challenged with 5 mg/kg body weight of LPS and observed survival. (**F**) Kaplan- Meier survival analysis. Data are pooled from three independent experiments. n = 14 for Ctrl mice; n = 11 for cKO mice. Inflammatory cytokines TNF-α (**G**) and MCP-1 (**H**) levels from mice serum before (0 h) and at 6 hr after (6 h) LPS injection are shown. n = 10 for Ctrl mice; n = 9 for cKO mice. Note greater than 500-fold increase in plasma TNF-α levels. (**I, J**) cKO mice mounted enhanced inflammatory response to acetaminophen-triggered liver necrosis. Serum TNF-α (**I**) and MCP-1(**J**) levels at 6 hr upon acetaminophen (3.2 mg/mouse) treatment are shown. n = 4 for Ctrl mice; n = 6 for cKO mice. Mann-Whitney test was used for statistics analysis, and lines indicate Mean ±SD. Similar trends were observed in another independent experiment.

DOI: https://doi.org/10.7554/eLife.32497.020

acetaminophen-triggered liver necrosis induces HMGB-1-mediated inflammatory responses to danger-associated molecular patterns (DAMPs) (*Chen et al., 2009*; *Scaffidi et al., 2002*). As shown in *Figure 8I* and *Figure 8J*, cKO mice mounted a significantly elevated inflammation to challenge by

low doses of acetaminophen. Therefore, amplification of IMLEC also leads to elevated response to tissue injuries.

Since *Mx1-Cre* was broadly activated after pIpC treatment, it is less certain whether the increased sensitivity of the cKO mice to TLR ligands is due to immunological abnormality. To address this issue, we produced chimeric mice in which pIpC induces deletion of the targeted gene exclusively in hematopoietic cells by transplanting *Rptor$^{F/F}$*, *Mx1-Cre* (CD45.2$^+$) BM cells into lethally irradiated CD45.1$^+$ recipients. After hematopoietic reconstitution, the recipients were treated with 3 doses of pIpC to induce deletion of *Rptor* exclusively in the hematopoietic cells. After 10 days of pause, the Ctrl and cKO chimera mice were challenged with new pIpC injection and monitored for survival (*Figure 9A*). As shown in *Figure 9B*, while a large portion of the cKO chimera mice died progressively starting within a week of the second round of pIpC treatment, all Ctrl chimera mice survived the observation period of more than 45 days. Massive leukocytes infiltration was observed in the liver of cKO chimeric mice (*Figure 9C*). Cell surface phenotyping of the donor-type leukocytes in BM (*Figure 9D*), spleen (*Figure 9E*) and liver (*Figure 9F*) revealed accumulation of CD11b$^+$ Gr-1$^-$ PD-L1$^+$ F4/80$^{low/-}$ IMLECs. Collectively, the data in *Figure 8* and *Figure 9* demonstrate that over-expansion of IMLECs, as a result of *Rptor* deletion, renders the host highly vulnerable to TLR ligands.

## Discussion

We have characterized a leukocyte population, which we called IMLECs, that has a strong innate effector function but with features of both myeloid and lymphoid cells. Furthermore, our data reveal an unexpected function of mTORC1 in suppressing IMLEC expansion. The high vulnerability to TLR ligands after IMLEC expansion highlights a new consequence of defective hematopoiesis and a new mechanism of immune tolerance.

Two groups have previous reported that inactivation of mTORC1 by deletion of *Rptor* leads to massive accumulation of CD11b$^+$Gr-1$^-$ cells in the BM (*Hoshii et al., 2012*; *Kalaitzidis et al., 2012*). Similar results were obtained in mice with *Mtor* deletion in the hematopoietic cells (*Guo et al., 2013*). We have demonstrated here that despite expression of a myeloid cell marker CD11b, the CD11b$^+$Gr-1$^-$ BM cells have lymphoid morphology. The active production of sterile transcripts at Ig loci suggests that these cells have partially committed to the B-cell lineage, while lack of VDJ rearrangement and cell surface B cell markers suggests that the differentiation toward the B cell lineage is limited. Importantly, this cell population expresses high levels of PD-L1 but does not express markers for other lymphoid cells including T cells (CD3) and NK cells (NK1.1) as well as for myeloid cells including F4/80, Gr-1 and CD115. Expression of CD11b also distinguishes IMLECs from ILCs (*Walker et al., 2013*), which are CD11b$^-$. PCA analysis demonstrated that IMLECs are distinct from but close to B cells and macrophages in gene expression profiles. While at much lower frequencies, cells with the same phenotypes and functional properties were also identified in normal BM, peripheral lymphoid and non-lymphoid organs.

It is of interest to note that the hallmark of IMLECs is the high expression of cell surface PD-L1. First identified as B7-H1 (*Dong et al., 1999*), PD-L1 has been shown to be involved in tumor evasion of T cell immunity, both by inducing exhaustion of effector T cells and by shielding tumor cells from effector T cells (*Hirano et al., 2005*; *Barber et al., 2006*). With the induction by cytokines such as IFN-γ and hypoxic tumor microenvironment, PD-L1 has been found on both tumor cells and host inflammatory cells such as myeloid derived DCs (*Curiel et al., 2003*; *Dong et al., 2002*), tumor-infiltrating myeloid derived suppressor cells (*Noman et al., 2014*). Recent studies have demonstrated that PD-L1 is an important biomarker and therapeutic target in cancer immunotherapy (*Garon et al., 2015*; *Brahmer et al., 2012*). Since IMLECs constitutively express high levels of PD-L1, it will be of interest to investigate their function in cancer immunity. It's worth noting that IMLECs of Raptor-deficient BM have an overall higher expression of PD-L1 than that of Raptor-sufficient BM, perhaps this reflects the indirect consequence of reduced mTORC1-mediated translation of PD-L1 negative regulators.

Since the IMLEC population expanded at the expense of granulocytes in the BM, we tested if IMLECs were derived by trans-differentiation of the granulocytes. Our genetic analyses demonstrated that inactivation of mTORC1 in the granulocytes, using *Lyz2-Cre*, failed to produce this subset, suggesting that loss of mTORC1 in granulocytes does not cause their trans-differentiation into IMLECs. Furthermore, since inactivation of mTORC1 in CD11b$^+$Gr-1$^+$ granulocytes did not affect

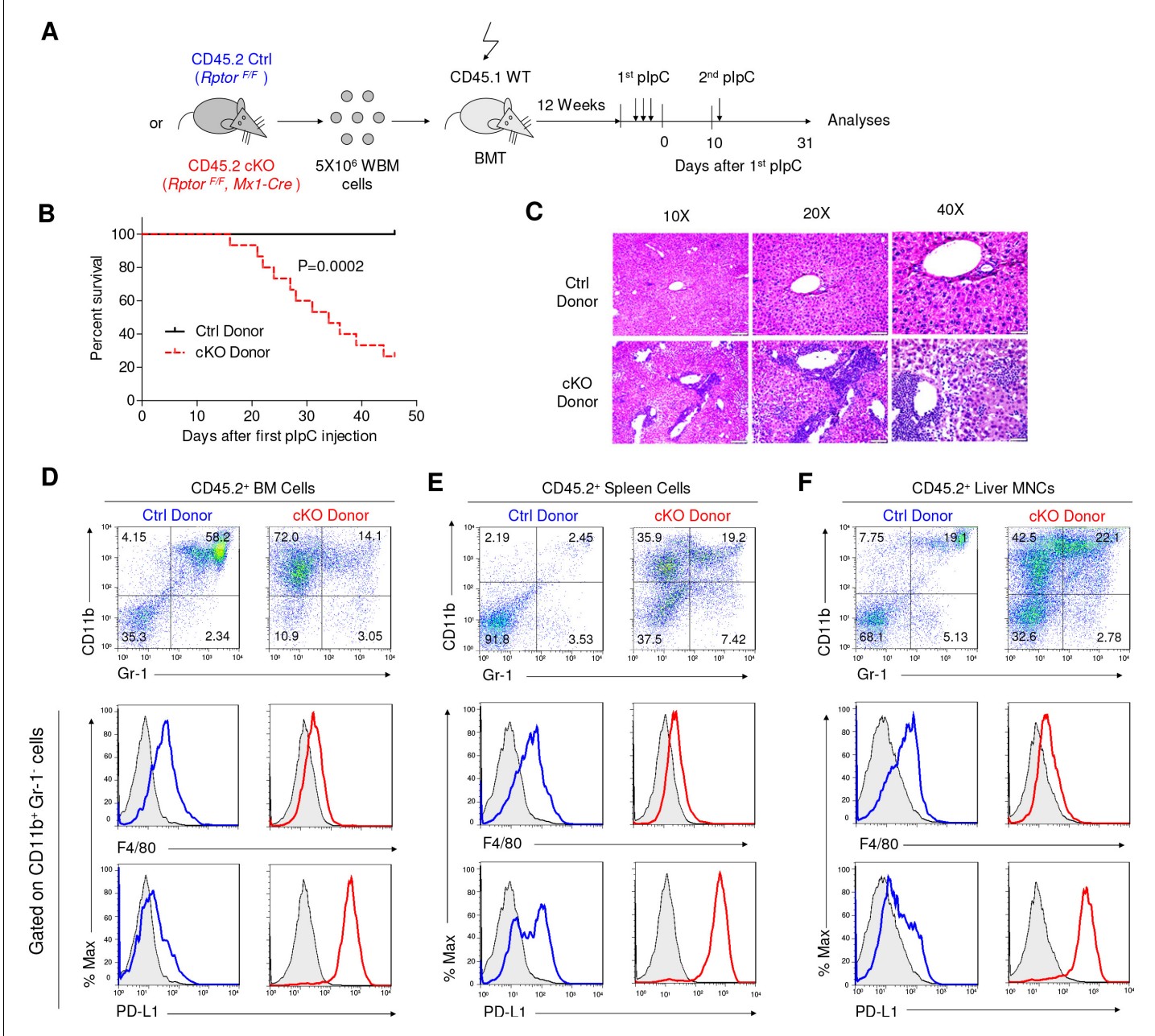

**Figure 9.** *Rptor* deletion in hematopoietic cells greatly increases vulnerability of mice to pIpC. (A) Diagram of experimental design. WT CD45.1[+] recipient mice were irradiated and transplanted with $5 \times 10^6$ BM cells from *Rptor^{F/F}* and *Rptor^{F/F}, Mx1-Cre* BM. After the fully reconstitution, recipients were treated with pIpC three times every other day to induce gene deletion in chimera mice with cKO BM. 10 days after the last injection of pIpC, recipients were challenged with another pIpC injection and the survival of mice were followed for four more weeks. Arrows denote the injections of pIpC at indicated time points. (B) Kaplan-Meier survival analysis after pIpC treatments. n = 12 for Ctrl chimera mice; n = 15 for cKO chimera mice. (C) Histological analysis of liver sections after H&E staining. Note extensive inflammation and liver damage in cKO chimera mice. Scale bars represent 200 μm (10X), 100 μm (20X) and 50 μm (40X). (D–F) Identification of IMLEC in BM (D), spleen (E) and livers (F) by flow cytometry. Mice were euthanized 45 days after the first pIpC injection. The distinct CD11b[+] Gr-1[-] PD-L1[+] F4/80[low/-] IMLECs greatly enriched in BM, spleen and liver of recipients of *Rptor* cKO BM. FACS profiles shown represent one of five independent experiments.

DOI: https://doi.org/10.7554/eLife.32497.021

their abundance in BM, loss of granulocytes in the mice with broad deletion of *Rptor* in all hematopoietic cells was not due to a cell-intrinsic requirement for mTORC1 in survival of granulocytes. Since IMLECs generally do not undergo active proliferation and are prone to apoptosis, their massive accumulation is most likely due to continuous production rather than self-renewal. Indeed, while LSKs and CMPs differentiate into IMLECs both in vitro and in vivo, IMLECs are not able to propagate in vivo (data not shown). Since *Lyz2-Cre* mediated *Rptor* deletion does not result in accumulation of IMLECs, it is obvious that IMLEC differentiation pathway is initiated before *Lyz2-Cre* expression which also occurs in some CMPs, and not all CMPs can give rise to IMLEC. On the other hand, since CLPs give rise to IMLEC in vitro, the possibility that other progenitor cells may give rise to IMLECs has not been ruled out.

Since heterozygous mutation of *Myb* causes an increase of CD11b$^+$Gr-1$^{low/-}$ cells in BM (*García et al., 2009*), we tested if conditional deletion of *Myb* is sufficient to cause accumulation of IMLEC. Our data demonstrate that cell-intrinsic reduction of *Myb* in BM resulted in accumulation of CD11b$^+$ Gr-1$^-$ PD-L1$^+$ IMLECs. It is of interest to note that mice with inactivation of *Rptor* share many hematopoietic phenotypes with mice harboring targeted disruption of *Myb*, such as an increase in HSPCs (*Sandberg et al., 2005*) and defective B-lymphopoiesis (*Fahl et al., 2009*; *Thomas et al., 2005*; *Greig et al., 2010*). Moreover, consistent with the proposed roles of c-Myb in regulating precise hematopoietic commitments, CMPs with deletion of either *Rptor* or *Myb* favored differentiation to CD11b$^+$ Gr-1$^-$ cells (*Lieu and Reddy, 2012*). Since our data show that the majority of the CD11b$^+$Gr-1$^-$ cells generated from CMPs are IMLECs, these data support the notion that IMLEC accumulation in *Rptor* cKO BM is caused by reduced *Myb* expression in CMPs.

While the mechanism by which mTORC1 down-regulates *Myb* remains to be fully elucidated, we have found a general up-regulation of putative miRNAs targeting *Myb* in HSPCs after *Rptor* deletion. Our data suggest that mTORC1 inactivation in HSPCs expands IMLECs by down-regulation of *Myb*, perhaps through increased miRNA biogenesis.

Previous studies by us and others have demonstrated a critical role for regulated mTOR signaling in hematopoiesis (*Yilmaz et al., 2006*; *Zhang et al., 2006*; *Chen et al., 2008*). Thus, while mTOR activation by deletion of *Tsc1* complex expands the numbers of hematopoietic stem cells, it causes reduced hematopoietic stem cell function but induced leukemic stem cells in conjunction with *Pten* deletion. This is consistent with the generally accepted association between defective differentiation and leukemiogenesis, which forms the foundation for treatment of leukemia through induction of differentiation (*Nowak et al., 2009*). The expansion of IMLECs caused by mTORC1 inactivation did not lead to leukemiogenesis as IMLECs are non-dividing cells that undergo a high rate of apoptosis. Instead, our data demonstrate a new consequence of defective differentiation, namely generation of a new population of cells with distinct effector function, as discussed below.

Our RNA-seq data suggest that IMLECs broadly over-express pattern recognition receptors for innate immunity. Corresponding to a broad, although not universal TLR elevation, IMLECs mount drastically exacerbated responses to all TLR ligands tested in vitro. Intracellular cytokine staining revealed that both normal and *Rptor*-cKO IMLECs were among the most active producers of inflammatory cytokines. Surprisingly, although *Rptor*$^{-/-}$ BM cells have normal TLR5 levels, they produce 10–20 folds more inflammatory cytokines, such as TNF-α and MCP-1 in response to TLR5 ligand. It is therefore likely that beyond TLRs, IMLECs have acquired other features that enhance their response to TLR ligands.

Apart from BM, expanded IMLECs are broadly distributed in lymphoid and non-lymphoid organs such as lung and liver. Our data demonstrate that an increase in IMLEC numbers makes the host highly vulnerable to TLR ligands such as LPS and polyI:C, as indicated by rapid demise of mice following pIpC and low doses of LPS injections. Since necrosis of normal tissues leads to release of endogenous TLR ligands, such as HMGB1 (*Scaffidi et al., 2002*) and HSP70 (*Millar et al., 2003*), and since unregulated host response to HMGB1 leads to fatal inflammation (*Chen et al., 2009*), it is conceivable that expansion of IMLECs may also make the host more vulnerable to tissue injury-induced inflammation. This is evidenced by elevated inflammation in acetaminophen-triggered liver necrosis model.

The severe pathological consequences may explain the paradoxical functions of mTOR in regulation of inflammation. While mTOR signaling is known to be activated by inflammatory cytokines (*Chen et al., 2010*) and rapamycin has been shown to inhibit production of inflammatory cytokines (*Abraham and Wiederrecht, 1996*), perhaps through its inhibition of mTOR-mediated NFκB

activation (*Lee et al., 2007*), rapamycin has been shown to induce inflammation in a small number of transplantation patients (*Diekmann et al., 2012*; *Thaunat et al., 2005*; *Dittrich et al., 2004*). Likewise, we have observed that rapamycin increases production of inflammatory cytokines in autoimmune Scurfy mice and in mice treated with high doses of endotoxin (*Chen et al., 2010*). It would be of great interest to determine if rapamycin can expand IMLECs in transplantation patients and in mice that either receive high doses of endotoxin or are genetically predisposed to autoimmune disease. It is also reported that persistent mTORC1 inhibition can result in elevated inflammation, activation of STAT3 and enhanced hepatocellular carcinoma development (*Umemura et al., 2014*). Our findings focused on characterizing IMLECs and inflammation support the provocative findings of this report, and might provide alternative and complementary explanations. However, despite an extensive effort, we have failed to induce IMLECs by mTOR kinase inhibitor Torin2 or rapamycin in WT mice (data not shown). Therefore, under normal circumstances, pharmaceutical inhibition of mTORC1 alone cannot achieve comparable levels of inflammation as those achieved by genetic inactivation of either *Mtor* or *Rptor*. Additional conditions must be met for mTOR inhibitors to cause inflammation. While these unknown barriers have ensured safety of mTOR inhibitors in most circumstances, their breakdown may explain paradoxical induction of inflammation by rapamycin.

The concept of immune tolerance has traditionally been reserved for adaptive immunity to avoid autoimmune diseases. A multitude of mechanisms, including clonal deletion (*Sha et al., 1988*; *Kisielow et al., 1988*; *Kappler et al., 1987*), clonal anergy (*Nossal and Pike, 1980*; *Schwartz et al., 1989*) and dominant regulatory T cells (*Sakaguchi et al., 1995*), have been described to reduce self-reactive T and B cell clone sizes to avoid autoimmune diseases. On the other hand, innate immunity is known to be regulated at levels of cellular activation (*Kärre et al., 1986*) and cellular recruitment (*Springer, 1994*). However, we and others have reported that innate immune effectors, especially NK cells, have features of adaptive immunity as their immune protective function against cancer and viruses is amplified through increased population sizes (*Gao et al., 2003*; *Sun et al., 2009*). However, a regulatory mechanism to control innate effector population size for the sake of preventing self-destruction has not been described. Our discovery of a developmentally regulated mechanism to control the population size of IMLECs to avoid unwanted self-destruction, as described herein, reveals a parallel between adaptive and innate immunity to avoid potentially life-threatening inflammation and tissue damage. It is therefore of interest to consider the concept of immune tolerance in the area of innate immunity.

An important consideration is whether IMLEC is a normal population of hematopoietic cells or a population that arises after pathogenic mutations. While we have identified cells of similar phenotypes and functional properties in normal mice, they are extremely rare and thus have no obvious physiological functions unless they are substantially expanded. We have demonstrated that these cells do not undergo proliferation and are prone to apoptosis, and that their expansion depends on abnormal hematopoiesis. Therefore, the pathological consequence observed herein is only known to manifest itself if the mTORC1-Myb pathway is genetically inactivated, resulting in disruption of normal hematopoiesis. Further studies are needed to identify conditions that can lead to accumulation of IMLEC short of these known mutations.

In summary, our data demonstrate that inactivation of mTORC1 in hematopoietic stem/progenitor cells leads to generation of IMLECs, a new cell population that shares features with myeloid and lymphoid lineages. The greater than 500-fold increase in population size of IMLECs in mTORC1- or *Myb*-defective BM highlights the critical role for mTORC1 and c-Myb in repressing the development of sufficient number of IMLECs to cause serious inflammation and tissue damage. Our study reveals a new consequence of defective hematopoiesis and may help to extend the concept of immune tolerance to innate immunity.

## Materials and methods

**Key resources table**

| Reagent type (species) or resource | Designation | Source or reference | Identifiers | Additional information |
|---|---|---|---|---|
| strain, strain background (*Mus musculus,C57BL/6*) | $Rptor^{F/F}$ | PMID: 19046572 | | |

*Continued on next page*

*Continued*

| Reagent type (species) or resource | Designation | Source or reference | Identifiers | Additional information |
|---|---|---|---|---|
| strain, strain background (*Mus musculus,C57BL/6*) | $Myb^{F/F}$; $Myb^{F/F},Rag2^{-/-},CreER$ | PMID: 15195090; PMID: 16169500 | | |
| strain, strain background (*Mus musculus,C57BL/6*) | Mx1-Cre | The Jackson Laboratory | Stock No: 003556; RRID:IMSR_JAX:003556 | |
| strain, strain background (*Mus musculus,C57BL/6*) | CreER | The Jackson Laboratory | Stock No: 007001; RRID:IMSR_JAX:007001 | |
| strain, strain background (*Mus musculus,C57BL/6*) | Lyz2-Cre | The Jackson Laboratory | Stock No: 004781; RRID:IMSR_JAX:004781 | |

## Mice and induction of gene deletion

$Rptor^{F/F}$ mice (*Bentzinger et al., 2008*) were crossed to the C57BL/6 background for more than 10 generations. $Myb^{F/F}$ and $Myb^{F/F}$, $Rag2^{-/-}$, CreER mice were reported previously (*Fahl et al., 2009*; *Thomas et al., 2005*; *Bender et al., 2004*). The interferon-inducible Mx1-Cre transgenic mice (RRID: IMSR_JAX:003556) (*Kühn et al., 1995*), tamoxifen-inducible Cre-ER$^{T2}$ transgenic mice (RRID:IMSR_JAX:007001) (*Indra et al., 1999*) and Lyz2-Cre knock-in mice (RRID:IMSR_JAX:004781) (*Clausen et al., 1999*) with C57BL/6 background were purchased from the Jackson Laboratory. $Rptor^{F/F}$ mice were crossed with Mx1-Cre mice, Lyz2-Cre mice or CreER mice to produce $Rptor^{F/F}$, Mx1-Cre (cKO) mice, $Rptor^{F/F}$, Lyz2-Cre$^{+/+}$ mice or $Rptor^{F/F}$, CreER mice, respectively. $Myb^{F/F}$ mice were crossed with Mx1-Cre mice to generate $Myb^{F/F}$ (Ctrl) and $Myb^{F/F}$, Mx1-Cre (cKO) mice. Offspring of these mice were genotyped by PCR-based assays with genomic DNA from mouse tail snips. Mice were cared for in the Unit of Laboratory Animal Medicine (ULAM) at the University of Michigan, where these studies were initiated, or Research Animal Facility (RAF) of Children's National Medical Center, where the studies were completed. All procedures involving experimental animals were approved by the University Committee on the Use and Care of Animals (UCUCA) at the University of Michigan or Children's National Medical Center.

Raptor Ctrl and cKO mice used in each experiment were sex-matched littermates. Mice were given 2 mg/kg body weight of pIpC (GE Healthcare Life Sciences) or 400 µg pIpC (Sigma-Aldrich) every other day for consecutive 3 to 7 times as specified by intra-peritoneal (i.p.) injection to induce Cre expression as in previous study (*Tang et al., 2012*). Deletion of target genes were confirmed as previously described (*Bentzinger et al., 2008*; *Bender et al., 2004*). Wild type C57BL/6 (CD45.2) mice and congenic C57BL/6 (CD45.1) mice were purchased from the Charles River Laboratories. Tamoxifen (Sigma-Aldrich) was dissolved in corn oil (Sigma-Aldrich) to 20 mg/ml and injected i.p. at 150 mg/kg/day for five consecutive days.

## Histology, cytology and complete blood cell count

Ctrl and cKO mice were euthanized by $CO_2$ inhalation on day 30 of last pIpC treatment. For histology, tissues were fixed in 10% neutral buffered formalin for 24–48 hr and the sternums were then decalcified in Immunocal (formic acid-based decalcifier, Decal, Tallman, NY) for 24 hr. Tissues were trimmed and cassetted and processed to wax on an automated processor using standard methods. Sections were cut at 5 µm thickness and hematoxylin and eosin-stained slides prepared on an automated stainer. For cytology, BM was collected from the femoral marrow cavity with a fine diameter paintbrush dipped in sterile PBS with 5% fetal bovine serum and cytology smears were prepared by gently brushing the collected cells in parallel lines on a glass slide. Cytology slides of BM smear and cytospins of FACS-sorted BM cells were stained using a Romanowsky-based stain (Diff-Quik, Hema 3 Manual staining system, Fisher Scientific).

Histological and cytological parameters were evaluated using an Olympus BX45 light microscope at total magnifications ranging from 40 X to 100 X (oil). Histological alterations were descriptively identified. Cytological alterations were descriptively identified and quantitative BM differential counts were made using a manual differential counter and standard criteria for cell identification. Images were taken using a 12.5 megapixel microscope-mounted Olympus DP72 digital camera and accompanying software (Olympus). Complete blood cell count was performed using the Hemavet 950 Hematology System (Drew Scientific Inc.) by the Animal Diagnostic Laboratory of ULAM Pathology Cores for Animal Research in the University of Michigan.

## Cells preparation and bleeding

BM cells were flushed out from the long bones (tibiae and femurs) by a 25-gauge needle with staining buffer (1XHanks Balanced Saline Solution without calcium or magnesium, supplemented with 2% heat-inactivated fetal bovine serum). Single cell suspensions of spleen, thymus, lung and lymph nodes were generated by gently squashing with frosted slides in a small volume of staining buffer. Cells from mouse peritoneal cavity were harvested as described before (*Ray and Dittel, 2010*). For isolation of mouse liver mononuclear cells, liver fragments were pressed through 70 μm round cell strainer (Becton Dickinson). Single-cell suspensions in a 35% Percoll solution (GE Healthcare) were centrifuged for 20 min at 800 g with brake off at room temperature. Pellet was collected and washed with staining buffer. Peripheral blood was collected by retro-orbital bleeding with heparinized capillary tubes or by submandibular bleeding with a lancet.

## Flow cytometry

For surface staining, cells were stained with the indicated antibodies (Abs) in staining buffer for 20 min at 4°C. In the characterization of surface markers for $CD11b^+$ $Gr-1^-$ cells, Fcγ receptors were pre-blocked by incubating cells with culture medium from hybridoma 2.4G2 (*Kurlander et al., 1984*) for 20 min at 4°C. For intracellular staining, cells were first stained with the indicated surface markers Abs and then fixed with Cytofix/Cytoperm buffer (BD Biosciences) for 1–2 hr at 4°C, followed by incubation with Cytoperm Plus buffer (BD Biosciences) for 15 min at room temperature (R.T.). After refixing for 15 min at R.T., cells were incubated with antibodies or isotype controls for 20 min (anti-TNF/IgG, anti-Ki-67Abs) or overnight (anti-c-Myb/IgG Abs) and further stained with the secondary Ab if necessary. BrdU labeling experiments were performed per the manufacture's instruction (BD Biosciences), as previously reported (*Chen et al., 2008*; *Tang et al., 2012*). Apoptosis assays by 7-AAD and Annexin V (BD Biosciences) were according to manufacturer's instructions. All FACS analyses were performed on a BD LSR II or a Canto II Flow Cytometer, and data were analyzed with FlowJo software (Tree Star, Inc.). The enrichment of $Lin^-$ BM cells was performed using MACS beads from mouse Lineage Cell Depletion Kit (Miltenyi Biotec). $CD11b^+$ $Gr-1^-$ BM cells from *Rptor* $^{F/F}$, *Mx1-Cre* mice, $Lin^-$ $CD11b^+$ $PD-L1^+$ IMLECs / LSK/CLP/ CMP populations from Raptor Ctrl/cKO or WT mice, $CD11b^+$ $Gr-1^+$ / $CD11b^-$ $Gr-1^-$ BM populations from *Rptor* $^{F/F}$, *Lyz2-Cre* $^{+/+}$ mice, $CD11b^{high}$ $Gr-1^-$ $F4/80^+$ peritoneal macrophages were sorted using FACSAria II or Influx cell sorter (BD Biosciences). The detailed information on Abs used in this study is in *Supplementary file 3*.

## BM cells transplantation

C57BL/6 Ly5.2 ($CD45.1^+$) recipient mice at the age of 6–12 weeks old were lethally irradiated for total 900–1,100 rads with a Cs-137 γ-ray source or a RS 2000 X-ray irradiator (Rad Source Technologies, Inc.). Indicated donor BM cells (whole BM or FACS-sorted BM cells) were transplanted into recipients through the mice tail by intra-venous (i.v.) injection within 24 hr after irradiation (*Tang et al., 2012*). At different time points post-transplantation, peripheral blood from the recipient mice was analyzed by flow cytometry to test the reconstitutions.

## Conventional PCR and quantitative PCR

Genomic DNA was isolated from BM cells by DNeasy Blood and Tissue Kit (Qiagen) as per manufacturer's instructions. Total RNA was isolated using TRIzol (Invitrogen) or ReliaPrep RNA Cell Miniprep System (Promega). Reverse transcription was carried out using random hemaxmer primers and SuperScript II Reverse Transcriptase (Invitrogen). Conventional PCR was performed using GoTaq Green Master Mix (Promega). Quantitative PCR (q-PCR) was performed by the 7500 real-time PCR system using Power SYBR Green Master Mixture (Applied Biosystems). Fold changes were calculated according to the ΔΔCT method (*Livak and Schmittgen, 2001*). The primers used for conventional PCR and q-PCR are listed in *Supplementary file 4*.

## Ig gene rearrangement test by V(D)J analysis

Immunoglobulin (Ig) gene recombination was determined using genomic DNA as previously described (*Riddell et al., 2014*). For Heavy chain, a semi-nested PCR strategy was employed to amplify the framework regions of VH to specific sites of JH. First round amplification of 25 cycles was performed with primers FR/JH1 (70°C annealing/20 s extension). Second round amplification of

35 cycles was with primers FR/JH2 (65°C annealing/30 s extension). Light chain (Igk and Igl) recombination was tested by primers $V_\kappa/J_\kappa 5$ and $V_\lambda 1/J_\lambda 1,3$ following previous report (Cobaleda et al., 2007). Sequences of primers used are listed in Supplementary file 4.

## OP9 cell co-culturing with HSPCs

OP9 stromal cell line (ATCC Cat# CRL-2749, RRID:CVCL_4398) was purchased from American Type Culture Collection (ATCC, Manassas, USA). No cell lines used in this study were listed in the database of cross-contaminated or misidentified cell lines suggested by International Cell Line Authentication Committee (ICLAC). All cell lines from ATCC were authenticated by the STR profiling method and tested as mycoplasma contamination free by ATCC. OP9 cells were maintained in α-MEM medium (Life Technologies) supplemented with 20% heat-inactivated fetal bovine serum (Hyclone), 100 units/ml of penicillin and 100 μg/ml of streptomycin (Gibco). The 6-well and 12-well flat-bottomed plates were pre-coated with OP9 cells at approximate 100% confluence after overnight growth. Subsequently $2 \times 10^3$ LSK (Lin⁻ c-Kit⁺ Sca-1⁺) or $5 \times 10^4$ CMP (Lin⁻ c-Kit⁺ Sca-1⁻ CD34$^{Medium}$CD16/32$^{Medium}$) cells or $5 \times 10^4$ CLP (Lin⁻ CD127⁺c-Kit$^{Medium}$ Sca-1$^{Medium}$) cells FACS-sorted from Raptor Ctrl/cKO BM were seeded. The co-culturing medium was additionally supplemented with 2 ng/ml murine recombinant IL-3, 2 ng/ml murine recombinant IL-6, 20 ng/ml murine recombinant SCF, 10 ng/ml murine recombinant Flt3L and 5 ng/ml murine recombinant IL-7 (all from R and D Systems). Lenti viral particles were produced in HEK 293 T cells (ATCC Cat# CRL-3216, RRID: CVCL_0063) by transiently co-transfecting control vector pWPI (Plasmid #12254, Addgene), or pWPI-Myb (cDNA of Myb was purchased from Dharmacon of GE Lifesciences, Catalog Number: MMM1013-202763262; Clone ID: 3672769) together with helper plasmids pMD2.G (Plasmid #12259, Addgene) and psPAX2 (Plasmid #12260, Addgene) using FuGENE HD Transfection Reagent (Promega). OP9 cells were replaced every 3–4 days by transferring co-culturing cells to new plates pre-coated with fresh OP9 cells. The hematopoietic cells in suspension were harvested on day 10–14 post seeding and subjected to flow cytometric analyses.

## TLR stimulation and inflammatory cytokine assay

For in vitro TLR stimulation, fresh BM cells were seeded in a 12-well plate with a density of $4 \times 10^6$ cells/ well(1 ml medium/well) or in a 48-well plate with a density of $1 \times 10^6$ cells/ well (200 μl medium/well). The culture medium was RPMI 1640 (Life Technologies) supplemented with 10% heat-inactivated fetal bovine serum (Hyclone), 100 units/ml of penicillin and 100 μg/ml of streptomycin (Gibco). BM cells were stimulated with 1 μg/ml LPS (Sigma-Aldrich, from O111:B4 E.coli) or a panel of TLR agonists (InvivoGen) for 16 hr. For intracellular cytokine staining, the protein transport inhibitor Brefeldin A (eBioscience) was added to the culturing medium during the last 4 hr of incubation.

The concentrations of the TLR agonists for in vitro studies were as following: TLR1/2-Pam3CSK4, 300 ng/ml; TLR2-HKLM, $10^8$ cells/ml; TLR3-pIpC(HMW), 10 μg/ml; TLR4-LPS-EK, 1 μg/ml; TLR5-FLA-ST, 1 μg/ml; TLR6/2-FSL-1, 100 ng/ml; TLR7-ssRNA-DR/LyoVec, 1 μg/ml; TLR9-ODN1826, 1 μM. For in vivo tests, pIpC (GE Healthcare, 2 mg/kg), LPS (from O55:B5 E.coli, Sigma-Aldrich, 5 mg/kg) and Acetaminophen (Children's TYLENOL, 3.2 mg/mouse) were injected i.p. Supernatant from in vitro cultured BM cells and serum from in vivo treated mice were assayed for inflammatory cytokines by BD Cytometric Bead Array (CBA)-Mouse Inflammation Kit according to the manufacturer's protocols.

## cDNA library preparation and RNA sequencing

FACS-sorted CD11b⁺ Gr-1⁻ BM cells from Raptor cKO mice and whole BM cells from Raptor Ctrl mice were used for RNA isolation with TRIzol Reagent (Life Technologies) per manufacturer's instructions. The cDNA libraries were constructed following the standard Illumina protocols by TruSeq RNA and DNA sample preparation kits (Illumina). Briefly, beads containing oligo (dT) were used to isolate poly(A) mRNA from total RNA. Purified mRNA was then fragmented in fragmentation buffer. Using these short fragments as templates, random hexamer-primers were used to synthesize the first-strand cDNA. The second-strand cDNA was synthesized using buffer, dNTPs, RNase H and DNA polymerase I. Short double-stranded cDNA fragments were purified for end repair and the addition of an 'A' base. Next, the short fragments were ligated to Illumina sequencing adaptors. DNA fragments of a selected size were gel-purified and amplified by PCR. The amplified cDNA libraries were

quality validated and then subjected to 50 nt single-end sequencing on an Illumina HiSeq 2000 at the University of Michigan DNA Sequencing Core.

## RNA-seq gene expression analysis

The reference sequences used were genome and transcriptome sequences downloaded from the UCSC website (version mm10). Clean reads were respectively aligned to the reference genome and transcriptome using Tophat (RRID:SCR_013035) (*Kim et al., 2013*). No more than two mismatches were allowed in the alignment for each read. Reads that could be uniquely mapped to a gene were used to calculate the expression level. The gene expression level was measured by the number of uniquely mapped reads per kilobase of exon region per million mappable reads (RPKM) and was calculated by DEGseq (RRID:SCR_008480) (*Wang et al., 2010*). The formula was defined as below:

$$\mathrm{RPKM} = \frac{10^6 \, \mathrm{C}}{\frac{\mathrm{NL}}{10^3}}$$

in which C was the number of reads uniquely mapped to the given gene; N was the number of reads uniquely mapped to all genes; L was the total length of exons from the given gene. For genes with more than one alternative transcript, the longest transcript was selected to calculate the RPKM.

## PCA, differential and cell-specific expression analysis

The RPKM method eliminates the influence of different gene lengths and sequencing discrepancies on the gene expression calculation. Therefore, the RPKM value can be used for comparing the differences in gene expression among samples. The RPKM value of all RNA-seq raw data were calculated according to the same workflow as stated above.

A function was implemented in the R software to perform principal component analysis (PCA). This function computes the eigenvalues and eigenvectors of the dataset (23498 genes) using the correlation matrix. The eigenvalues were then ordered from highest to lowest, indicating their relative contribution to the structure of the data. The projection of each sample defined by components was represented as a dot plot to generate the PCA figures.

Selected samples were then pooled by subtypes and a two sided t-test with FDR (False discovery rate) of 0.05 and fold change of 4 was performed to identify differentially expressed genes between IMLECs and other subtypes (mean RPKM values of genes in two subtypes both below five were deleted). For subtype-specific genes identified, a one-sided t-test (null hypothesis is greater) was performed with FDR of 0.01 and fold change of 4 contrasting each subtype in turn versus all other subtypes pooled, and statistically significant genes were assigned to the respective subtype signature.

RNA-seq datasets in this study have been deposited in the Gene Expression Omnibus (GEO) database as accession number GSE67863. Other public RNA-seq datasets used are as followings: peritoneal CD11b⁺F4/80⁺ macrophages (GSM1103013, GSM1103014 in GEO Series GSE45358); normal BM CD11b⁺ Gr-1⁺ granulocytes (GSM1166354, GSM1166355, GSM1166356 in GEO Series GSE48048); BM-derived dendritic cells (GSM1012795, GSM1012796 and GSM1012797 in GEO Series GSE41265); BM erythroid cells (GSM1208164, GSM1208165 and GSM1208166 in GEO Series GSE49843); BM pro B and pre B cells (GSM978778 and GSM978779 in GEO Series GSE39756); Naïve B cells (GSM1155172, GSM1155176, GSM1155180 and GSM1155184 in GEO Series GSE47703); activated B cells (GSM1155170, GSM1155174, GSM1155178 and GSM1155182 in GEO Series GSE47703); CD4 T cells, CD8 T cells and natural regulatory T (nTreg) cells (GSM1169492, GSM1169501, GSM1169493, GSM1169502, GSM1169499 and GSM1169508 in GEO Series GSE48138);. Spleen NK cells (GSM1257953, GSM1257954, GSM1257955 and GSM 1257956 in GEO Series GSE52047); Blood neutrophils (GSM1340629, GSM1340630, GSM1340631 and GSM1340632 in GEO Series GSE55633); Lung basophils and eosinophils (GSM1358432, GSM1358433, GSM1358436 and GSM1358437 in GEO Series GSE56292). RNA-seq data on ex vivo DC subsets are obtained from GEO Series GSE62704: GSM1531794 (CDP), GSM1531795 (pDC), GSM1531796 (preDC), GSM1531797 (DN DC), GSM1531798 (CD4⁺ DC), GSM1531799 (CD8⁺ DC).

## Genome browser display of immunoglobulin expression

To facilitate the global viewing of transcript structure and gene expression quantity of immunoglobulin, an interface in which RNA-seq gene expression of immunoglobulin can be viewed in Genome Browser display was constructed. For viewing and analysis, the UCSC Genome Browser (http://www.genome.ucsc.edu, RRID:SCR_005780) (*Kent et al., 2002*) with the mm10 version of the mouse genome was used. For each base in each cell type, the normalized number of aligned reads count was defined as below:

$$\text{Normalized number of aligned reads count} = A \star \frac{\sum_{k=0}^{n} N_k}{n}$$

in which N was the number of reads uniquely mapped to the given base in each cell type; n was the number of replicates for each cell type samples; A was a constant t in which takes the value of 3.5E + 7. The normalized number of aligned reads count for each base in each cell type was stored as WIG file and was uploaded to UCSC genome browser. For each track, y-axis depicted the normalized number of aligned reads count and x-axis depicts physical distance in bases along the chromosome.

## Statistical analysis

All the data are presented as mean ±SD. Unless otherwise indicated, two-tailed, unpaired student's *t* tests were used for comparison between two experimental groups. In *Figure 8I* and *Figure 8J*, where the data do not follow normal distribution, Mann-Whitney tests were used. The log-rank tests were used for the Kaplan-Meier survival analysis. Statistical significance was determined as p<0.05 (*p<0.05; **p<0.01; ***p<0.001).

## Acknowledgements

We thank Dr. Hans Schreiber (University of Chicago) and all the colleagues in our laboratory for their helpful discussions, and Ms. Morgan E Daley for editorial assistance. This work was supported by the grants from National Institute of Health AG036690, AI64350, CA183030 and CA171972. This work was initiated at the University of Michigan then completed at Children's Research Institute of Children's National Medical Center.

## Additional information

### Funding

| Funder | Grant reference number | Author |
| --- | --- | --- |
| National Institute of Allergy and Infectious Diseases | AI64350 | Yang Liu<br>Pan Zheng |
| National Cancer Institute | CA183030 | Yang Liu |
| National Institute on Aging | AG036690 | Pan Zheng |
| National Cancer Institute | CA171972 | Yang Liu |

The funders had no role in study design, data collection and interpretation, or the decision to submit the work for publication.

### Author contributions

Fei Tang, Conceptualization, Data curation, Formal analysis, Investigation, Methodology, Writing—original draft; Peng Zhang, Resources, Data curation, Software, Formal analysis, Validation, Investigation; Peiying Ye, Data curation, Investigation, Methodology; Christopher A Lazarski, Data curation, Methodology; Qi Wu, Ingrid L Bergin, Methodology; Timothy P Bender, Michael N Hall, Resources; Ya Cui, Liguo Zhang, Taijiao Jiang, Resources, Methodology; Yang Liu, Conceptualization, Formal analysis, Supervision, Funding acquisition, Validation, Visualization, Project administration, Writing—review and editing; Pan Zheng, Conceptualization, Data curation, Formal analysis, Supervision,

Funding acquisition, Validation, Investigation, Visualization, Methodology, Project administration, Writing—review and editing

## Author ORCIDs
Fei Tang (iD) http://orcid.org/0000-0003-1475-5028
Peng Zhang (iD) http://orcid.org/0000-0002-6218-1885
Yang Liu (iD) http://orcid.org/0000-0002-9442-700X
Pan Zheng (iD) http://orcid.org/0000-0003-2598-3544

## Ethics

Animal experimentation: This study was performed in strict accordance with the recommendations in the Guide for the Care and Use of Laboratory Animals of the National Institutes of Health. All of the animals were handled according to approved institutional animal care and use committee (IACUC) protocols (312-13-12 and #00030574) of the Children's National Medical Center. Every effort was made to minimize suffering.

## Decision letter and Author response

Decision letter https://doi.org/10.7554/eLife.32497.054
Author response https://doi.org/10.7554/eLife.32497.055

## Additional files

### Supplementary files

• Supplementary file 1. Transcript levels of CD markers analyzed from RNA-seq datasets. The FPKM values of 3 mice samples for cKO-IMLEC group (*Rptor* cKO CD11b$^+$Gr-1$^-$ BM IMLECs) and Ctrl-WBM group (Ctrl whole BM cells) are shown. Fold change indicates the ratio of average FPKM values.
DOI: https://doi.org/10.7554/eLife.32497.022

• Supplementary file 2. Expression levels of miRNAs targeting *Myb* in Raptor Ctrl and cKO BM HSPCs. The normalized signal values of 3 mice samples for cKO-HSPCs group and Ctrl-HSPCs group are shown. The microarray data have been deposited to NCBI GEO database with accession number GSE64042 as previously reported (*Ye et al., 2015*).
DOI: https://doi.org/10.7554/eLife.32497.023

• Supplementary file 3. The list of antibodies used for flow cytometry.
DOI: https://doi.org/10.7554/eLife.32497.024

• Supplementary file 4. Sequences of DNA primers used in PCR assays.
DOI: https://doi.org/10.7554/eLife.32497.025

• Transparent reporting form
DOI: https://doi.org/10.7554/eLife.32497.026

### Major datasets

The following dataset was generated:

| Author(s) | Year | Dataset title | Dataset URL | Database, license, and accessibility information |
|---|---|---|---|---|
| Tang F, Zhang P, Cui Y, Zhang L, Jiang T, Liu Y, Zheng P | 2015 | A lineage of myelolymphoblastic innate cells unmasked by inactivation of mTOR complex | http://www.ncbi.nlm.nih.gov/geo/query/acc.cgi?acc=GSE67863 | Publicly available at the NCBI Gene Expression Omnibus (accession no: GSE67863) |

The following previously published datasets were used:

| Author(s) | Year | Dataset title | Dataset URL | Database, license, and accessibility information |
|---|---|---|---|---|
| Behrendt R, Schumann T, Dahl A, Alexopoulou D | 2013 | Gene expression analysis of murine peritoneal macrophages deficient for SAMHD1 and IFNAR | http://www.ncbi.nlm.nih.gov/geo/query/acc.cgi?acc=GSE45358 | Publicly available at the NCBI Gene Expression Omnibus (accession no: GSE45358) |
| Ren M, Cowell JK | 2013 | Gene expression profiling normal murine myeloid cells and myeloid leukemia cells induced by CNTRL-FGFR1 | http://www.ncbi.nlm.nih.gov/geo/query/acc.cgi?acc=GSE48048 | Publicly available at the NCBI Gene Expression Omnibus (accession no: GSE48048) |
| Shalek AK, Satija R, Park H, Regev A | 2013 | Single-cell transcriptomics reveals widespread heterogeneity in gene regulation and RNA processing in stimulated immune cells | http://www.ncbi.nlm.nih.gov/geo/query/acc.cgi?acc=GSE41265 | Publicly available at the NCBI Gene Expression Omnibus (accession no: GSE41265) |
| Hosoya T, Engel JD | 2013 | TRIM28 is essential for erythroblast differentiation in the mouse | http://www.ncbi.nlm.nih.gov/geo/query/acc.cgi?acc=GSE49843 | Publicly available at the NCBI Gene Expression Omnibus (accession no: GSE49843) |
| Li P, Spolski R, Liao W, Wang L, Murphy TL, Murphy KM, Leonard WJ | 2012 | BATF-JUN is critical for IRF4-mediated transcription in T cells | http://www.ncbi.nlm.nih.gov/geo/query/acc.cgi?acc=GSE39756 | Publicly available at the NCBI Gene Expression Omnibus (accession no: GSE39756) |
| Hogenbirk MA, Jacobs H | 2013 | Differential programming of B cells in Aicda-/- mice | http://www.ncbi.nlm.nih.gov/geo/query/acc.cgi?acc=GSE47703 | Publicly available at the NCBI Gene Expression Omnibus (accession no: GSE47703) |
| Gangqing H, Qingsong T, Suveena S, Fang Y, Thelma E, Stefan M, Jinfang Z, Keji Z | 2013 | Expression and regulation of lincRNAs during T cell development and differentiation | http://www.ncbi.nlm.nih.gov/geo/query/acc.cgi?acc=GSE48138 | Publicly available at the NCBI Gene Expression Omnibus (accession no: GSE48138) |
| Artyomov MN | 2014 | Tissue-Resident Natural Killer (NK) Cells Are Cell Lineages Distinct From Thymic and Conventional Splenic NK Cells | http://www.ncbi.nlm.nih.gov/geo/query/acc.cgi?acc=GSE52047 | Publicly available at the NCBI Gene Expression Omnibus (accession no: GSE52047) |
| Coffelt SB, de Visser K | 2014 | Gene expression profile of metastasis-associated neutrophils | http://www.ncbi.nlm.nih.gov/geo/query/acc.cgi?acc=GSE55633 | Publicly available at the NCBI Gene Expression Omnibus (accession no: GSE55633) |
| Motomura Y, Kubo M | 2014 | Expression profiling by high throughput sequencing | http://www.ncbi.nlm.nih.gov/geo/query/acc.cgi?acc=GSE56292 | Publicly available at the NCBI Gene Expression Omnibus (accession no: GSE56292) |
| Bornstein C, Winter D, Barnett-Itzhaki Z, David E, Kadri S, Garber M, Amit I | 2014 | A negative feedback loop of transcription factors specifies alternative dendritic cell chromatin states | http://www.ncbi.nlm.nih.gov/geo/query/acc.cgi?acc=GSE62704 | Publicly available at the NCBI Gene Expression Omnibus (accession no: GSE62704) |
| Ye P, Liu Y, Chen C, Tang F, Wu Q, Wang X, Liu C, Liu X, Liu R, Zheng P | 2014 | mTORC1 regulates microRNA biogenesis in mouse bone marrow hematopoietic stem and progenitor cells | http://www.ncbi.nlm.nih.gov/geo/query/acc.cgi?acc=GSE64042 | Publicly available at the NCBI Gene Expression Omnibus (accession no: GSE64042) |

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
