## [Decision Letter]

[Editors’ note: a previous version of this study was rejected after peer review, but the authors submitted for reconsideration. The first decision letter after peer review is shown below.]

Thank you for submitting your work entitled "A Population of Innate Myelolymphoblastoid Effector Cell Expanded by Inactivation of mTOR Complex 1" for consideration by *eLife*. Your article has been reviewed by three peer reviewers, one of whom is a member of our Board of Reviewing Editors and the evaluation has been overseen by Tadatsugu Taniguchi as the Senior Editor. Our decision has been reached after consultation between the reviewers. Based on these discussions and the individual reviews below, we regret to inform you that your work will not be considered further for publication in *eLife*.

The following individual involved in the review of your submission has agreed to reveal his identity: (peer reviewer: Frederic Geissman).

While your study was thought to be interesting and thought-provoking by all reviewers, you will see that they all share major concerns about the characterization of WT 'IMLECs', and feel that without far more analysis and characterization of the so-called WT IMLECs, the report presented here does not warrant a case for them in normal myelopoiesis. The extent of additional work involved to address this concern would be far too much to be practical within a two-month revision window.

The reviewers also think that your demonstration of the 'Raptor-deletion-induced IMECs' and their potential role in pathogenic myeloid differentiation, particularly in specific situations of inflammatory pathology such as in patients with mutations in relevant pathways or in situations of long-term MTORC1 inhibition therapy, could be very interesting independent of any claims about identification of 'WT IMLECs'. However, as you will see from the reviews, they remain concerned that the present manuscript does not provide adequate clarity about the inflammatory impact of these specific cells in such situations and about the mechanisms involved. Of course, such a manuscript would be quite different from the present one, and will be considered as a new submission subject to a fresh and completely independent review process.

Reviewer #1:

The manuscript shows very interesting data demonstrating the characteristics of an odd innate myelo-lymphoid cell population in the normal mouse bone marrow that becomes very prominent in the bone marrow of the Raptor-deficient mouse, along with some data regarding the genesis and the functional consequences of this population. These are putatively novel findings and well worth publication. However, enthusiasm for publication would be enhanced if major concerns as identified below could be substantively addressed.

1) The manuscript focuses far too much on the issue of the role of cell numbers in regulating self-tolerance, as well as on the so-called 'novelty' of this cell population. While the first question is interesting, it is quite peripheral to the findings in the manuscript. The second claim of novelty somewhat over-interprets the data, which currently address the characterisation, genesis and functions of an identifiable group of leukocytes without actually quite identifying a stable differentiated cell lineage with normal physiological role/s.

2) The use of polyIC as an inducer of Cre-mediated gene deletion introduces a substantial potential confounder in an analysis of a putative myeloid cell subset. It would be advisable to have confirmatory data for most critical findings using another system of inducible Cre expression, such as the ER-Cre the authors have used.

3) The claim of the putative 'IMLECs' being present in wild-type bone marrow needs somewhat more robust support. Thus, the flow cytometric data showing such cells in normal bone marrow indicate two distinct populations in the gate, one CD11b^dull^PDL1^bright^ and one CD11b^bright^PDL1^dull^, neither of which precisely corresponds to the population in Raptor-deficient bone marrow cells. Upon differentiation in vitro, too, WT LSKs and CMPs generate apparent 'IMLECs' that show much lower levels of PD-L1 than Raptor-deficient 'IMLECs' do. Also, the data on TLR/PRR and cytokine expression on 'IMLECs' from WT bone marrow again show that these cells are PDL1low compared to 'IMLECs' from Raptor-deficient mice, and that they are less efficient at producing TNF-α. Greater clarity in the similarities and differences in this cell population from wild-type versus Raptor-deficient mice, both in terms of cell-surface phenotype and transcript profiles, is needed.

4) The finding that LysM-Cre-mediated rptor deletion does not lead to the generation of the putative IMLECs needs further explanation, since a substantial proportion of CMPs would be expected to express LysM-Cre and would therefore be expected to be redirected into an IMLEC differentiation pathway per the authors' model. It is thus possible that altered commitment is programmed prior to the CMP stage. This is further complicated by the fact that, while mixed chimeras using whole bone marrow cells generate Raptor-deficient myeloid cells that are mostly 'IMLECs' rather than granulocytic cells, this is not true when LSKs are used; Raptor-deficient LSK cells give rise mostly to granulocytic cells in vivo. It is possible that this differentiation pathway is imprinted as a result of the absence of Raptor in a very early hematopoietic cell stage leading to some degree of loss of myeloid/lymphoid distinction as well as downstream loss of granulocytic differentiation. Acknowledgement and some explanation of these ambiguities are essential.

5) The c-Myb-deletion data require further support. The phenotype of the heterozygous c-Myb-deleted mouse with reference to PDL1 expression needs to be shown. Also, the connection between c-myb and Raptor deficiencies in generating these putative IMLECs requires clarity; thus, for example, does provision of heterologous c-myb modify the generation of 'IMLECs' from Raptor-deficient LSK cells in vitro?

6) The claim that the explanation for the disease phenotype in vivo in Raptor-deficient mice solely or even primarily lies in the abnormally large numbers of normally functioning highly sensitive pro-inflammatory 'IMLECs' needs much more substantiation. At the very least, mixed bone marrow chimeras with graded contributions from Raptor-deficient and WT bone marrow need to be done to examine the disease consequences when Raptor-deficient 'IMLEC' cell numbers approach their apparent WT counterparts.

Reviewer #2:

Research article – Tang et al. A population of Innate Myelolymphoblastoid Effector Cell Expanded by Inactivation of mTOR Complex 1

The authors report a previously uncharacterized leukocyte subset (Lin-, CD11b+, Gr1-, PD-L1+) expressing IgH and Rag2, they name IMLEC. IMLEC are present in the bone marrow of Rptor deficient, pIpc treated mice, as well, in much lower numbers in the BM of Rptor ^F/F^, pIpc treated mice. The authors also show the presence of cells with similar features in the spleen BM and blood of wt mice. When genetic deletion of *Rptor* or *Myb* is introduced in Mx1-Cre expressing cells the population expands dramatically, in a cell-autonomous manner. IMLEC are also present in the progeny of Myb-deficient hematopoietic progenitors. IMLEC produce cytokines in response to TLR stimulation, and accumulate in tissues. The authors conclude that mTOR is able to suppress IMLEC expansion via *Myb* (in a relatively mysterious manner), and when mTOR is dysregulated, IMLEC production can lead to severe pathological consequences due to a high vulnerability of IMLECs to TLR ligands.

The experiments are well performed, the authors analyze IMLECs in depth and describe how genetic dysregulation of hematopoietic cells can lead to the aberrant expansion of these dysfunctional myelo/lymphoid cells. The main interrogation about this study, is whether this cell type is physiological, or only appears in the absence of Rptor or Myb. The characterization of 'WT' IMLEC is not convincing. Nevertheless, this study indicates how cell autonomous genetic events in hematopoietic progenitors can lead to inflammation, by the generation of 'abnormal' myeloid cells.

Altogether, this study is interesting and well performed, but the description of a new physiological cell type is not convincing, and the manuscript could be revised to focus on the 'pathological' consequences of Raptor deficiency.

The claim of the paper describing WT IMLEC as a cell type is not convincing, and extensive experiments would be needed if the authors were to maintain this claim.

This work is however very interesting as there was evidence in particular in a recent article in Cell by the group of M. Karin that 'long term mTORC1 inhibition' (in this model raptor deletion, or rapamycin treatment) in vivo induces inflammation, and this pathway is of obvious relevance as longterm rapamycin treatment is used or maybe(?) misused in human. The authors observations may support the provocative findings of Karin, and provide an (alternative) or complementary mechanism for this important observation.

Specific comments:

Figure 1: It is, just by looking at the HE stain, not possible to appreciate the expansion of the lymphocyte population. Controls would need to be displayed bigger. (see also comment for Figure 1—figure supplement 1

Figure 1—figure supplement 1: Display control spleen in the same way as the Rptor-/- spleen (comparable to liver in Figure 8 where control and experimental animals are shown in the same magnifications)

Figure 1—figure supplement 4: ST-HSCs are CD150- and CD48-, please revise the FACS plots and the quantification in B

Figure 3: Sorted wild-type IMLECs do not look exactly as Rptor-deficient IMLECs (shown in Figure 1). This brings back the concern mentioned above of whether these cells are the same cell type, or whether the KO changes the phenotypic nature of WT IMLECs. The authors should at least sort and stain Ctrl and KO cells at the same time to see whether these histological differences are not due to technical reasons. It would be also of interest to see these cells in comparison in histograms showing their FSC and SSC sizes as shown in Figure 1—figure supplement 2.

Figure 3: The authors speculate about the CD11chigh population in the liver, which could be CD8^+^ DCs. Since these are WT samples, it would be reasonable to repeat this experiment and include CD8 in the lineage or to define these populations further.

Figure 5: The authors should either (always) speculate that miRNAs might be the reason for Myb downregulation in Rptor^-/-^ mice, or (always) strongly state it. For now, it seems that the authors could not decide as their statements change in the text and the figure legends.

Reviewer #3:

Tang et al. described here a previously reported population of cell expanded in a mouse model presenting an inactivation of the mTOR Complex 1. These cells present a lymphoid morphology with a myeloid phenotype, clustering with macrophages at the gene expression level. They also express a wide range of TLR and are functional, secreting pro-inflammatory cytokines when challenged.

The main concern here is that the nature of the cell population remains elusive especially in the WT environment: is it a real independent lineage? Is it a "blocked and frustrated" progenitor stage? Most importantly, its physiological relevance is also not addressed since this population is seen mostly when deleting Rptor in hematopoietic stem/progenitor cells and as its existence in WT bone marrow is really not convincingly addressed. It rather appears as a blocked and dysfunctional dead end of abnormal hematopoiesis that does not exist in normal hematopoiesis. Again, the authors do not provide enough convincing data supporting the significance of this population.

In term of phenotype, the authors found that IMLEC express PD-L1 and CD11b. Do they express other markers such as Ly6C? MHC-II? Of note, CD11b is indeed a myeloid marker but also express by lymphoid cells such as NK cells and cannot really be used as a lineage marker. In addition, among the 317 CD markers described in Supplementary file 1, the authors should test other markers than PDL1. Also, since these markers where found from the KO IMLEC, any risk that some of the negative markers tested are controlled by functional mTOR complex? Phenotype should be tested in WT bone marrow.

The authors used RNAseq data to investigate the nature of the cells. However, they isolated IMLEC from Rptor deficient bone marrow and hence, the lack of functional mTOR could blur the true nature of the cells. The authors should perform gene expression profiling using WT cells. Also, it is not clear in the comparison of gene expression profiles of KO IMLEC with other known subsets of hematopoietic cells if the authors sorted the cells for the other known subsets from KO, WT mice or use only public database. If the latter, they should precise if all these arrays were made with a comparable approach using similar aged and sex match mice of the same background.

The characterization of the WT IMLEC is not convincing at all and the data provided mostly correlative. How comparable are the WT and KO populations in phenotype, gene expression profile, TLR expression and function.

The statement "Perhaps the CD11c^high^ cells in the spleen are the CD8^+^ PDL-1^+^ DCs and the CD11c^low/-^ cells are IMLECs." is not acceptable. The authors should prove it.

To identify the progenitor that may give rise to IMLECs, the authors omitted to test the contribution of CLP, which will be important to test as CLP give rise to B cells and IMLEC present high levels of sterile transcripts thay may suggest that they correspond to a stage before early B cell commitment.

How do the authors exclude that the broad expression of TLR could be a response to induction of Mx1 by pIpC administration?

For the Ki67 and annexin stainings, the authors are comparing IMLEC to the rest of the bone marrow, which is a mix of heterogeneous population of progenitors and differentiated cells. This is not really comparable and not relevant. For proliferation, the authors should rather do a Brdu labeling or cell cycle assay by flow cytometry that measure DNA content (and which is more accurate and sensitive than Ki67 staining).

[Editors’ note: what now follows is the decision letter after the authors submitted for further consideration.]

Thank you for submitting your article "A Population of Innate Myelolymphoblastoid Effector Cell Expanded by Inactivation of mTOR Complex 1" for consideration by *eLife*. Your article has been reviewed by three peer reviewers, one of whom is a member of our Board of Reviewing Editors and the evaluation has been overseen by Tadatsugu Taniguchi as the Senior Editor. The reviewers have opted to remain anonymous.

The reviewers have discussed the reviews with one another and the Reviewing Editor has drafted this decision to help you prepare a revised submission.

Summary:

The manuscript is an extensively revised re-submission of an earlier version showing very interesting data demonstrating the characteristics of an odd innate myelo-lymphoid cell population in the normal mouse bone marrow that becomes very prominent in the bone marrow of the Raptor-deficient mouse, along with some data regarding the genesis and the functional consequences of this population. These are putatively novel findings and well worth publication. The authors have made efforts to address any, though not all, of the concerns expressed regarding the earlier version. Some reservations do, however, remain.

Essential revisions:

1) Specifically, it is still not quite clear if the data indicate a stable differentiated cell lineage with normal physiological role/s, and/or a specific non-redundant role in the inflammatory phenotype seen in Rptor-deficient mice. The characterisation of the IMLECs in WT mice remains preliminary and is not fully convincing, making it unclear if they really exist as a homogeneous population in normal bone marrow. The massive generation of IMLEC following Myb deletion (revised Figure 6) further and strongly suggests that IMLEC may be some kind of a preleukemic cell type. It is not clear if WT IMLECs preserve their phenotype stably in vivo, and if induced deletion of Rptor in them lead to further, putatively aberrant differentiation. Therefore, the manuscript needs to acknowledge these issues and substantially modulate the claims to identification of a new 'normal' bone marrow cell type in a far more modest and qualified direction.

2) More convincingly, the study indicates how cell autonomous genetic events in hematopoietic progenitors can lead to inflammation by the generation of 'abnormal' myeloid cells. That issue, too, will be strengthened by demonstrations of the disease-related consequences of WT and/or Rptor-deficient IMLECs.

[Editors' note: further revisions were requested prior to acceptance, as described below.]

Thank you for resubmitting your work entitled "A Population of Innate Myelolymphoblastoid Effector Cell Expanded by Inactivation of mTOR Complex 1 in Mice" for further consideration at *eLife*. Your revised article has been favorably evaluated by Tadatsugu Taniguchi (Senior editor), a Reviewing editor.

The manuscript has been improved but there are some remaining issues that need to be addressed before acceptance, as outlined below:

The authors have addressed the revisions advised in the earlier decision to some extent. However, it is essential to make further major and extensive text modifications (1) to reflect the primary focus of the manuscript on the disease-related role/s of Rptor-deficient IMLECs, and (2) to make it much more clear that the provenance of the IMLEC-like population found in WT mice is quite unclear at the moment.

---

## [Author Response]

[Editors’ note: the author responses to the first round of peer review follow.]

Reviewer #1:The manuscript shows very interesting data demonstrating the characteristics of an odd innate myelo-lymphoid cell population in the normal mouse bone marrow that becomes very prominent in the bone marrow of the Raptor-deficient mouse, along with some data regarding the genesis and the functional consequences of this population. These are putatively novel findings and well worth publication. However, enthusiasm for publication would be enhanced if major concerns as identified below could be substantively addressed.

We appreciate the expert evaluation and recommendation of publication of our work. We performed feasible experiments and made corresponding figure/text amendment to address the major concerns, as detailed below:

1) The manuscript focuses far too much on the issue of the role of cell numbers in regulating self-tolerance, as well as on the so-called 'novelty' of this cell population. While the first question is interesting, it is quite peripheral to the findings in the manuscript. The second claim of novelty somewhat over-interprets the data, which currently address the characterisation, genesis and functions of an identifiable group of leukocytes without actually quite identifying a stable differentiated cell lineage with normal physiological role/s.

We appreciate the comments and performed additional experiments to extensively characterize the corresponding population in the wild type mouse. Although with an extremely low abundance, the WT IMLECs were proven to display same morphology, size and granularity, expression for sterile transcripts of Ig loci (Figure 4), comparable levels of TLR expression (Figure 7), much similar extents of specific transcription factors expression (Figure 4), comparable ability in TNF production (Figure 7). These data demonstrated that IMLEC existed under physiological condition at low numbers and thus support the notion that Raptor deletion increases IMLEC population size rather than creates a new cell population. As such, we believe it is reasonable to state that the new function of Raptor described herein is to prevent expansion of IMLEC under physiological conditions, and in doing so, it prevents pathological consequences associated with IMLEC expansion.

We would like to cast the new findings in the context of adaptive and innate immune regulatory mechanisms. However, we do understand the reviewer’s preference, and carefully revised the text to reflect the reviewer’s advice.

2) The use of polyIC as an inducer of Cre-mediated gene deletion introduces a substantial potential confounder in an analysis of a putative myeloid cell subset. It would be advisable to have confirmatory data for most critical findings using another system of inducible Cre expression, such as the ER-Cre the authors have used.

Rptor^F/F^, CreER mice were generated. Same phenotypes were observed in BM cells from both regular Rptor^F/F,^ CreER mice and recipient mice with Rptor^F/F^, CreER BM transplantation, after tamoxifen induced Rptor deletion (Figure 1—figure supplement 5).

3) The claim of the putative 'IMLECs' being present in wild-type bone marrow needs somewhat more robust support. Thus, the flow cytometric data showing such cells in normal bone marrow indicate two distinct populations in the gate, one CD11b^dull^PDL1^bright^ and one CD11b^bright^PDL1^dull^, neither of which precisely corresponds to the population in Raptor-deficient bone marrow cells. Upon differentiation in vitro, too, WT LSKs and CMPs generate apparent 'IMLECs' that show much lower levels of PDL1 than Raptor-deficient 'IMLECs' do. Also, the data on TLR/PRR and cytokine expression on 'IMLECs' from WT bone marrow again show that these cells are PDL1low compared to 'IMLECs' from Raptor-deficient mice, and that they are less efficient at producing TNF-α. Greater clarity in the similarities and differences in this cell population from wild-type versus Raptor-deficient mice, both in terms of cell-surface phenotype and transcript profiles, is needed.

Additional experiments were performed to more extensively characterize IMLECs from wild type mice (Figure 1—figure supplement 5; Figure 4, Figure 4, Figure 4; Figure 4—figure supplement 1; Figure 7). In toto, we presented evidence for IMLEC in WT mice from the following aspects: (i) Morphology (Giemsa staining, Figure 4); (ii) Size (FSC) and granularity(SSC) (Figure 4); (iii) expression for sterile transcripts of Ig loci (qPCR, Figure 4); (iv) More surface markers other than revealed before (flow cytometry, Figure 4—figure supplement 1); (v) TLRs expression (qPCR, Figure 7); (vi) Population specific transcriptional factors (qPCR, Figure 4); (vii) Apoptosis (Annexin A/7-AAD staining) and Proliferation (Ki-67/BrdU incorporation)(Figure 4—figure supplement 1); (viii) TNF production in vitro (intracellular flow staining, Figure 7). Further clarification of the similarities and differences (such as PD-L1 expression intensity) between IMLECs originated from WT and Raptor-deficient mice BM was stated in the current manuscript.

4) The finding that LysM-Cre-mediated rptor deletion does not lead to the generation of the putative IMLECs needs further explanation, since a substantial proportion of CMPs would be expected to express LysM-Cre and would therefore be expected to be redirected into an IMLEC differentiation pathway per the authors' model. It is thus possible that altered commitment is programmed prior to the CMP stage. This is further complicated by the fact that, while mixed chimeras using whole bone marrow cells generate Raptor-deficient myeloid cells that are mostly 'IMLECs' rather than granulocytic cells, this is not true when LSKs are used; Raptor-deficient LSK cells give rise mostly to granulocytic cells in vivo. It is possible that this differentiation pathway is imprinted as a result of the absence of Raptor in a very early hematopoietic cell stage leading to some degree of loss of myeloid/lymphoid distinction as well as downstream loss of granulocytic differentiation. Acknowledgement and some explanation of these ambiguities are essential.

Since LSK also contained LT-HSC and ST-HSC, the difference between whole bone marrow and LSK does not necessarily suggest IMLEC comes from earlier progenitors. Rather, we think the generation of granulocytes from LSK relates to the short-term transplantation protocol designed to study progenitors. LSK was used as positive control.

Transplantation of progenitor cells in the context of induced deletion is difficult because the progenitor cells do not self-renew and thus could further differentiate before the deletion effect. We believe some, although most likely not all, CMP can generate IMLEC based on in vitro co-culture and in vivo transplantation studies. This does not exclude other possibilities, as detailed in revised texts.

5) The c-Myb-deletion data require further support. The phenotype of the heterozygous c-Myb-deleted mouse with reference to PDL1 expression needs to be shown. Also, the connection between c-Myb and Raptor deficiencies in generating these putative IMLECs requires clarity; thus, for example, does provision of heterologous c-Myb modify the generation of 'IMLECs' from Raptor-deficient LSK cells in vitro?

Mice with heterozygous c-Myb deletion (Myb^F/+^, Mx1-Cre) did not show significant accumulation of IMLECs (Figure 6—figure supplement 1). However, deletion of c-Myb in mice with homozygous floxed c-Myb (Myb^F/F^, Mx1-Cre) did show obvious increase of PD-L1 expression in CD11b^+^Gr-1^-^ BM cells (Figure 6, Figure 6, Figure 6). The provision of heterologous c-Myb significantly diminished the generation of IMLECs from Raptor-deficient LSK cells in our in vitro OP9 co-culturing experiments (Figure 6—figure supplement 1).

6) The claim that the explanation for the disease phenotype in vivo in Raptor-deficient mice solely or even primarily lies in the abnormally large numbers of normally functioning highly sensitive pro-inflammatory 'IMLECs' needs much more substantiation. At the very least, mixed bone marrow chimeras with graded contributions from Raptor-deficient and WT bone marrow need to be done to examine the disease consequences when Raptor-deficient 'IMLEC' cell numbers approach their apparent WT counterparts.

Even when more mutant BM is provided in competitive reconstitution assays (Figure 5 and Figure 5, and also unpublished data) and others (*Cell Stem Cell*, 2012: 11, 429–439), the Raptor-deficient hematopoietic stem cells are drastically inferior to Raptor-sufficient hematopoietic stem cells in giving rise to progenies in mixed BM chimeras. The WT BM-derived cells will be overwhelmingly dominant and the contribution of Rptor^-/-^ IMLECs cannot be evaluated in this mixed BM model. We also attempted to transfer Rptor^-/-^ IMLECs into Rag1^-/-^ mice but these cells did not survive in vivo. As a result, we were not able use adoptive transfer to substantiate pathogenicity of IMLEC.

On the other hand, we have presented data that IMLEC produce high levels of cytokines which are known to be pathogenic in mice.

We have now carefully discussed this issue to reflect the reviewer’s concern.

Reviewer #2:The authors report a previously uncharacterized leukocyte subset (Lin-, CD11b+, Gr1-, PD-L1+) expressing IgH and Rag2, they name IMLEC. IMLEC are present in the bone marrow of Rptor deficient, pIpc treated mice, as well, in much lower numbers in the BM of Rptor ^f/f^, pIpc treated mice. The authors also show the presence of cells with similar features in the spleen BM and blood of wt mice. When genetic deletion of Rptor or Myb is introduced in Mx1-Cre expressing cells the population expands dramatically, in a cell-autonomous manner. IMLEC are also present in the progeny of Myb-deficient hematopoietic progenitors. IMLEC produce cytokines in response to TLR stimulation, and accumulate in tissues. The authors conclude that mTOR is able to suppress IMLEC expansion via Myb (in a relatively mysterious manner), and when mTOR is dysregulated, IMLEC production can lead to severe pathological consequences due to a high vulnerability of IMLECs to TLR ligands.

We appreciate the expert evaluation of our work.

The experiments are well performed, the authors analyze IMLECs in depth and describe how genetic dysregulation of hematopoietic cells can lead to the aberrant expansion of these dysfunctional myelo/lymphoid cells. The main interrogation about this study, is whether this cell type is physiological, or only appears in the absence of Rptor or Myb. The characterization of 'WT' IMLEC is not convincing. Nevertheless, this study indicates how cell autonomous genetic events in hematopoietic progenitors can lead to inflammation, by the generation of 'abnormal' myeloid cells.Altogether, this study is interesting and well performed, but the description of a new physiological cell type is not convincing, and the manuscript could be revised to focus on the 'pathological' consequences of Raptor deficiency.

Additional experiments were performed to more extensively characterize IMLECs from wild type mice (Figure 1—figure supplement 5; Figure 4, Figure 4, Figure 4; Figure 4—figure supplement 1; Figure 7). In toto, we presented evidence for IMLEC in WT mice from the following aspects: (i) Morphology (Giemsa staining, Figure 4); (ii) Size (FSC) and granularity(SSC) (Figure 4); (iii) expression for sterile transcripts of Ig loci (qPCR, Figure 4); (iv) More surface markers other than revealed before (flow cytometry, Figure 4—figure supplement 1); (v) TLRs expression (qPCR, Figure 7); (vi) Population specific transcriptional factors (qPCR, Figure 4); (vii) Apoptosis (Annexin A/7-AAD staining) and Proliferation (Ki-67/BrdU incorporation)(Figure 4—figure supplement 1); (viii) TNF production in vitro(intracellular flow staining, Figure 7). Further clarification of the similarities and differences (such as PD-L1 expression intensity) between IMLECs originated from WT and Raptor-deficient mice BM was stated in the current manuscript.

The claim of the paper describing WT IMLEC as a cell type is not convincing, and extensive experiments would be needed if the authors were to maintain this claim.This work is however very interesting as there was evidence in particular in a recent article in Cell by the group of M. Karin that 'long term mTORC1 inhibition' (in this model raptor deletion, or rapamycin treatment) in vivo induces inflammation, and this pathway is of obvious relevance as longterm rapamycin treatment is used or maybe(?) misused in human. The authors observations may support the provocative findings of Karin, and provide an (alternative) or complementary mechanism for this important observation.

The work from Michael Karin et al. was cited and discussed in our current manuscript.

As stated in the manuscript, we have failed to induce IMLECs by mTOR kinase inhibitor Torin2 or rapamycin in WT mice, despite an extensive effort. Therefore, under normal circumstances, pharmaceutical inhibition of mTORC1 alone cannot achieve comparable levels of inflammation. Probably additional conditions must be met for mTOR inhibitors to cause inflammation. However, our findings focused on characterizing IMLECs and inflammation support the provocative findings of Michael Karin *et al.* and might provide alternative and complementary explanations.

Specific comments:Figure 1: It is, just by looking at the HE stain, not possible to appreciate the expansion of the lymphocyte population. Controls would need to be displayed bigger. (see also comment for Figure 1—figure supplement 1Figure 1—figure supplement 1: Display control spleen in the same way as the Rptor^-/-^ spleen (comparable to liver in Figure 8 where control and experimental animals are shown in the same magnifications)

Control staining is now in the same magnification as that of cKO in Figure 1 and Figure 1—figure supplement 1.

Figure 1—figure supplement 4: ST-HSCs are CD150- and CD48-, please revise the FACS plots and the quantification in B

ST-HSCs are CD150^+^ and CD48^+^, as defined from the publications on SLAM receptors and HSC (*Cell Stem Cell*. 2013 Jul 3;13(1):102-16); *Cell*. 2005 Jul 1;121(7):1109-21), as well as evidenced from previous publication from our laboratory (*J Exp Med.* 2008 Sep 29; 205(10): 2397–2408.).

Figure 3: Sorted wild-type IMLECs do not look exactly as Rptor-deficient IMLECs (shown in Figure 1). This brings back the concern mentioned above of whether these cells are the same cell type, or whether the KO changes the phenotypic nature of WT IMLECs. The authors should at least sort and stain Ctrl and KO cells at the same time to see whether these histological differences are not due to technical reasons. It would be also of interest to see these cells in comparison in histograms showing their FSC and SSC sizes as shown in Figure 1—figure supplement 2.

We performed additional studies as advised. Ctrl and cKO IMLECs showed almost identical morphology according to Giemsa staining (Figure 4). They also display comparable size and granularity assayed by flow cytometry (Figure 4).

Figure 3: The authors speculate about the CD11c^high^ population in the liver, which could be CD8^+^ DCs. Since these are WT samples, it would be reasonable to repeat this experiment and include CD8 in the lineage or to define these populations further.

We removed the statements on speculating CD8^+^ DC from the manuscript.

Figure 5: The authors should either (always) speculate that miRNAs might be the reason for Myb downregulation in Rptor^-/-^ mice, or (always) strongly state it. For now, it seems that the authors could not decide as their statements change in the text and the figure legends.

After revision, the tune on the statement of miRNA and Myb downregulation is consistent in the text and the figure legend in the current manuscript.

Reviewer #3:Tang et al. described here a previously reported population of cell expanded in a mouse model presenting an inactivation of the mTOR Complex 1. These cells present a lymphoid morphology with a myeloid phenotype, clustering with macrophages at the gene expression level. They also express a wide range of TLR and are functional, secreting pro-inflammatory cytokines when challenged.The main concern here is that the nature of the cell population remains elusive especially in the WT environment: is it a real independent lineage? Is it a "blocked and frustrated" progenitor stage? Most importantly, its physiological relevance is also not addressed since this population is seen mostly when deleting Rptor in hematopoietic stem/progenitor cells and as its existence in WT bone marrow is really not convincingly addressed. It rather appears as a blocked and dysfunctional dead end of abnormal hematopoiesis that does not exist in normal hematopoiesis. Again, the authors do not provide enough convincing data supporting the significance of this population.

We appreciate the reviewer’s constructive criticism of our work. Additional experiments were performed to more extensively characterize IMLECs from wild type mice (Figure 1—figure supplement 5; Figure 4, Figure 4, Figure 4; Figure 4—figure supplement 1; Figure 7). In toto, we presented evidence for IMLEC in WT mice from the following aspects: (i) Morphology (Giemsa staining, Figure 4); (ii) Size (FSC) and granularity(SSC) (Figure 4); (iii) expression for sterile transcripts of Ig loci (qPCR, Figure 4); (iv) More surface markers other than revealed before (flow cytometry, Figure 4—figure supplement 1); (v) TLRs expression (qPCR, Figure 7); (vi) Population specific transcriptional factors (qPCR, Figure 4); (vii) Apoptosis (Annexin A/7-AAD staining) and Proliferation (Ki-67/BrdU incorporation)(Figure 4—figure supplement 1); (viii) TNF production in vitro (intracellular flow staining, Figure 7). Further clarification of the similarities and differences (such as PD-L1 expression intensity) between IMLECs originated from WT and Raptor-deficient mice BM was stated in the current manuscript

In term of phenotype, the authors found that IMLEC express PDL1 and CD11b. Do they express other markers such as Ly6C? MHC-II? Of note, CD11b is indeed a myeloid marker but also express by lymphoid cells such as NK cells and cannot really be used as a lineage marker. In addition, among the 317 CD markers described in Supplementary file 1, the authors should test other markers than PDL1. Also, since these markers where found from the KO IMLEC, any risk that some of the negative markers tested are controlled by functional mTOR complex? Phenotype should be tested in WT bone marrow.

The surface markers including MHC-I, MHC-II and Ly6C were evaluated by flow cytometry (Figure 4—figure supplement 1). IMLECs are positive for MHC-I and MHC-II, largely negative/low for Ly6C. Among the 317 CD markers, other surface markers (CD14, CD68) with commercially available antibodies were also tested. Since the tested top-high markers have largely comparable expression levels between WT and Rptor^-/-^ IMLECs, the chance for them to be the negatively regulated targets of mTORC1 is low. However, we cannot exclude the possibility that some other candidate markers are really negative markers controlled by mTORC1.

The authors used RNAseq data to investigate the nature of the cells. However, they isolated IMLEC from Rptor deficient bone marrow and hence, the lack of functional mTOR could blur the true nature of the cells. The authors should perform gene expression profiling using WT cells. Also, it is not clear in the comparison of gene expression profiles of KO IMLEC with other known subsets of hematopoietic cells if the authors sorted the cells for the other known subsets from KO, WT mice or use only public database. If the latter, they should precise if all these arrays were made with a comparable approach using similar aged and sex match mice of the same background.

We used the RNA-seq datasets from CD11b^+^Gr-1^-^ Rptor-deficient IMLECs for mining of potential surface markers, transcription factors to further investigate the nature of this population. Once the key factors are identified, we performed targeted analyses to compare WT and mutant IMLEC. Further validating of WT IMLECS in terms of gene expression profile was performed by comparing the surface markers expression (Figure 4—figure supplement 1), TLR genes expression (Figure 7) and population-specific transcription factors (Figure 4, Figure 4).

In addition, as stated in the Materials and methods section, public datasets of various known hematopoietic populations were used for comparison with cKO IMLECS. These datasets were from largely similar age of wild type C57BL/6 mice and compared with a comparable approach. Actually, publicly available datasets were widely used for comparative analysis and integrating to generate new results by researchers worldwide, such as these published papers: *Immunity*. 2014 Sep 18; 41(3):465-77; *Nat Methods*. 2013 Aug; 10 (8):795-803; *Genome Biol.* 2008 Jan 24; 9(1):R17.

The characterization of the WT IMLEC is not convincing at all and the data provided mostly correlative. How comparable are the WT and KO populations in phenotype, gene expression profile, TLR expression and function.

Additional experiments were performed to more extensively characterize IMLECs from wild type mice (Figure 4, Figure 4, Figure 4; Figure 4—figure supplement 1; Figure 7).

The statement " Perhaps the CD11chigh cells in the spleen are the CD8^+^ PDL-1^+^ DCs and the CD11c^low/-^ cells are IMLECs." is not acceptable. The authors should prove it.

CD11c^high^ population in the spleen is clearly dentritic cell population, while IMLECs are CD11c^low/-^. Since dendritic cells comprise a wide range of heterogeneous populations of cells, in order to avoid confusions on defining IMLECs, we removed the statements on speculating CD8^+^ DC from the manuscript.

To identify the progenitor that may give rise to IMLECs, the authors omitted to test the contribution of CLP, which will be important to test as CLP give rise to B cells and IMLEC present high levels of sterile transcripts thay may suggest that they correspond to a stage before early B cell commitment.

The contribution of CLPs in giving rise to IMLECs was evaluated (Figure 5, Figure 5). Our data demonstrate that the CLPs have the potential to give rise to IMLECs. We have modified to text to reflect the reviewer’s concern.

How do the authors exclude that the broad expression of TLR could be a response to induction of Mx1 by pIpC administration?

Another inducible deletion system (Rptor^F/F^, CreER mice) was introduced. Same phenotypes were observed in BM cells from both regular Rptor^F/F^, CreER mice and recipient mice with Rptor^F/F^, CreER BM transplantation, after tamoxifen induced Rptor deletion (Figure 1—figure supplement 5).

For the Ki67 and annexin stainings, the authors are comparing IMLEC to the rest of the bone marrow, which is a mix of heterogeneous population of progenitors and differentiated cells. This is not really comparable and not relevant. For proliferation, the authors should rather do a Brdu labeling or cell cycle assay by flow cytometry that measure DNA content (and which is more accurate and sensitive than Ki67 staining).

Comparisons between IMLECs and other defined lineages were undertaken in terms of apoptosis (Figure 4—figure supplement 1, Figure 4—figure supplement 1) and proliferation assayed by BrdU incorporation (Figure 4—figure supplement 1, Figure 4—figure supplement 1).

[Editors' note: the author responses to the re-review follow.]

The reviewers have discussed the reviews with one another and the Reviewing Editor has drafted this decision to help you prepare a revised submission.

We would like to take this opportunity again to thank the editors and reviewers for their constructive suggestions. We have made necessary changes of text to further acknowledge some issues and concerns from the editors and reviewers, as well as to fully meet the requirements of *eLife* publishing policies. As a result, we believe that the manuscript is greatly strengthened and ready for publication.

Summary:The manuscript is an extensively revised re-submission of an earlier version showing very interesting data demonstrating the characteristics of an odd innate myelo-lymphoid cell population in the normal mouse bone marrow that becomes very prominent in the bone marrow of the Raptor-deficient mouse, along with some data regarding the genesis and the functional consequences of this population. These are putatively novel findings and well worth publication. The authors have made efforts to address any, though not all, of the concerns expressed regarding the earlier version. Some reservations do, however, remain.

We appreciate the expert evaluation and recommendation of publication of our work.

Essential revisions:1) Specifically, it is still not quite clear if the data indicate a stable differentiated cell lineage with normal physiological role/s, and/or a specific non-redundant role in the inflammatory phenotype seen in Rptor-deficient mice. The characterisation of the IMLECs in WT mice remains preliminary and is not fully convincing, making it unclear if they really exist as a homogeneous population in normal bone marrow. The massive generation of IMLEC following Myb deletion (revised Figure 6) further and strongly suggests that IMLEC may be some kind of a preleukemic cell type. It is not clear if WT IMLECs preserve their phenotype stably in vivo, and if induced deletion of Rptor in them lead to further, putatively aberrant differentiation. Therefore, the manuscript needs to acknowledge these issues and substantially modulate the claims to identification of a new 'normal' bone marrow cell type in a far more modest and qualified direction.

We agree with the suggestions on further acknowledging these issues, and we have now carefully discussed these issues to reflect the editors and reviewers’ concerns.

2) More convincingly, the study indicates how cell autonomous genetic events in hematopoietic progenitors can lead to inflammation by the generation of 'abnormal' myeloid cells. That issue, too, will be strengthened by demonstrations of the disease-related consequences of WT and/or Rptor-deficient IMLECs.

We appreciate the constructive suggestion on strengthening the significance of work by revealing the disease-related consequences of IMLEC population. We have demonstrated the association between the expansion of Rptor-deficient IMLECs and lethal inflammatory responses to TLR ligands in mouse (Figure 7, Figure 8 and Figure 9). This is the first report on uncovering the critical innate immune functions of the CD3^-^B220^-^NK1.1^-^Ter119^-^ CD11c^low/-^CD115^-^F4/80^low/-^Gr-1^-^ CD11b^+^PD-L1^+^ population in mouse, and to the best of our knowledge, there is no related clinical report on human IMLECs till now. On the other hand, since RPTOR and MTOR are embryonically lethal genes, it is almost impossible to identify naturally occurring immune cells with completely abrogated mTORC1 activity and patients with genetically predisposed RPTOR/MTOR mutations.

[Editors' note: further revisions were requested prior to acceptance, as described below.]

Thank you for resubmitting your work entitled "A Population of Innate Myelolymphoblastoid Effector Cell Expanded by Inactivation of mTOR Complex 1 in Mice" for further consideration at eLife. Your revised article has been favorably evaluated by Tadatsugu Taniguchi (Senior editor), a Reviewing editor.The manuscript has been improved but there are some remaining issues that need to be addressed before acceptance, as outlined below:The authors have addressed the revisions advised in the earlier decision to some extent. However, it is essential to make further major and extensive text modifications (1) to reflect the primary focus of the manuscript on the disease-related role/s of Rptor-deficient IMLECs, and (2) to make it much more clear that the provenance of the IMLEC-like population found in WT mice is quite unclear at the moment.

Many thanks for your recent decision and for sending further comments of the reviewers. We have now included the following paragraph in the Discussion section in order to address the reviewer’s concern.

“An important consideration is whether IMLEC is a normal population of hematopoietic cells or a population that arises after pathogenic mutations. […] Further studies are needed to identify conditions that can lead to accumulation of IMLEC short of these known mutations.”

We hope you will find these modifications satisfactory.